# Genome evolution and diversity of wild and cultivated potatoes

Dié Tang[1,8], Yuxin Jia[1,8], Jinzhe Zhang[2,8], Hongbo Li[1,3,8], Lin Cheng[1], Pei Wang[1], Zhigui Bao[1], Zhihong Liu[1], Shuangshuang Feng[2], Xijian Zhu[4], Dawei Li[1], Guangtao Zhu[4], Hongru Wang[5], Yao Zhou[1], Yongfeng Zhou[1], Glenn J. Bryan[6], C. Robin Buell[7], Chunzhi Zhang[1] & Sanwen Huang[1✉]

Potato (*Solanum tuberosum* L.) is the world's most important non-cereal food crop, and the vast majority of commercially grown cultivars are highly heterozygous tetraploids. Advances in diploid hybrid breeding based on true seeds have the potential to revolutionize future potato breeding and production[1–4]. So far, relatively few studies have examined the genome evolution and diversity of wild and cultivated landrace potatoes, which limits the application of their diversity in potato breeding. Here we assemble 44 high-quality diploid potato genomes from 24 wild and 20 cultivated accessions that are representative of *Solanum* section *Petota*, the tuber-bearing clade, as well as 2 genomes from the neighbouring section, *Etuberosum*. Extensive discordance of phylogenomic relationships suggests the complexity of potato evolution. We find that the potato genome substantially expanded its repertoire of disease-resistance genes when compared with closely related seed-propagated solanaceous crops, indicative of the effect of tuber-based propagation strategies on the evolution of the potato genome. We discover a transcription factor that determines tuber identity and interacts with the mobile tuberization inductive signal SP6A. We also identify 561,433 high-confidence structural variants and construct a map of large inversions, which provides insights for improving inbred lines and precluding potential linkage drag, as exemplified by a 5.8-Mb inversion that is associated with carotenoid content in tubers. This study will accelerate hybrid potato breeding and enrich our understanding of the evolution and biology of potato as a global staple food crop.

Potato (*Solanum tuberosum* L.) belongs to the *Petota* section of the *Solanum* genus within the Solanaceae family, which contains many economically important species[5]. The *Petota* section consists of more than 100 tuber-bearing species, and is sister to the non-tuber-bearing *Etuberosum* section and the *Lycopersicon* section that comprises tomato species[5]. Commercial production of potato is dominated by autotetraploid cultivars that are propagated using seed tubers. Reinventing potato from a clonally propagated tetraploid to a true seed-propagated diploid has the potential to considerably accelerate genetic improvement, and would enable the genome design of a crop that has been highly recalcitrant to the use of molecular breeding and genomics approaches[3,6,7]. Diploid potatoes represent around 70% of the wild and landrace potato species[5], and the vast diversity among them has not been fully characterized or made use of in previous breeding programs. Furthermore, the effects of the evolution of a clonal reproduction strategy on potato genomes and the evolutionary mechanisms of tuberization are largely unexplored. So far, several potato genome sequences have been released, which have been important resources for genetics and breeding[3,8–13]. However, the minor portion of biodiversity in the *Petota* section that is captured by these genomes is insufficient to obtain a comprehensive understanding of the potato genome and tuber evolution. Here we report genome sequences and analyses of 44 diploid potatoes, as well as 2 species in the *Etuberosum* section. Our findings provide insights into the alteration of potato genomes during the evolution of tuberization, and will enable genome design for new diploid hybrids.

## Pan-genome of the *Petota* section

To capture the genome diversity of the *Petota* section, we selected 44 representative accessions based on the phylogenetic relationships of 432 accessions[7,14,15] (Supplementary Fig. 1). These comprise 20

[1]Shenzhen Branch, Guangdong Laboratory of Lingnan Modern Agriculture, Genome Analysis Laboratory of the Ministry of Agriculture and Rural Affairs, Agricultural Genomics Institute at Shenzhen, Chinese Academy of Agricultural Sciences, Shenzhen, China. [2]Key Laboratory of Biology and Genetic Improvement of Horticultural Crops of the Ministry of Agriculture, Sino-Dutch Joint Laboratory of Horticultural Genomics, Institute of Vegetables and Flowers, Chinese Academy of Agricultural Sciences, Beijing, China. [3]Graduate School Experimental Plant Sciences, Laboratory of Plant Breeding, Wageningen University and Research, Wageningen, The Netherlands. [4]The AGISCAAS-YNNU Joint Academy of Potato Sciences, Yunnan Normal University, Kunming, China. [5]Department of Integrative Biology, University of California Berkeley, Berkeley, CA, USA. [6]Cell and Molecular Sciences, The James Hutton Institute, Invergowrie, UK. [7]Center for Applied Genetic Technologies, University of Georgia, Athens, GA, USA. [8]These authors contributed equally: Dié Tang, Yuxin Jia, Jinzhe Zhang, Hongbo Li. ✉e-mail: huangsanwen@caas.cn

landraces, covering 5 indigenous cultivated diploid groups (landrace), 4 accessions from *Solanum candolleanum* (CND), which is considered the progenitor of cultivated potatoes, and another 20 wild potato species (4 from clades 1 and 2; 16 from clades 3 and 4, as defined in a previous study[5]) (Supplementary Table 1). We generated an average of 24.5 Gb (approximately 30-fold relative to the estimated haploid potato genome size of around 800 Mb) high-fidelity (HiFi) reads for the 44 accessions (Supplementary Table 1); these were de-novo-assembled into raw assembled contigs with heterozygous regions retained and into monoploid assembled contigs (MTGs), with average N50 contig sizes of 9.10 Mb and 23.33 Mb, respectively (Extended Data Fig. 1a,b, Extended Data Fig. 2, Supplementary Figs. 2 and 3 and Supplementary Table 1). Among these, seven representative genomes were assembled to chromosome level using high-throughput chromatin conformation capture (Hi-C)[16,17] sequencing data (Supplementary Fig. 4). The raw assembly size ranged from 835.1 Mb (A6-26) to 1.71 Gb (PG6246) (Extended Data Fig. 1a); this is positively correlated to the estimated heterozygosity, which was determined using $k$-mer-based methods ($R^2 = 0.47$, $P = 2.5 \times 10^{-7}$) (Extended Data Fig. 1c). The completeness of assemblies was supported by BUSCO[18], with an average score of 96.58% (single-copy and duplicated) in raw assembled contigs and 96.12% in MTGs (Supplementary Table 1). We predicted 44,859 (A6-26) to 88,871 (PG6002) gene models by integrating transcriptome evidence, homology-based prediction and ab initio prediction (Supplementary Table 1).

To build a comprehensive gene repertoire within the *Petota* section, we constructed a pan-genome by clustering the 2,701,787 predicted gene models from the 44 accessions and the reference genome of *S. tuberosum* Group Phureja (accession DM1-3 516 R44; hereafter referred to as DM)[8,11] into 51,401 pan-gene clusters using the Markov clustering algorithm[19]. Pan-genome size increased when incorporating more genomes and nearly reached a plateau when $n$ was close to 40 (Extended Data Fig. 1d), which suggests that our panel captures the shared gene content of potato. We next classified these clusters into four categories based on their frequency of occurrence: core clusters (present in all 45 accessions; 13,123; 25.5%), soft-core clusters (present in 42–44 accessions; 5,743; 11.2%), shell clusters (found in 2–41 individuals; 28,471; 55.4%) and accession-specific clusters (4,064; 7.9%) (Extended Data Fig. 1d, Supplementary Table 2 and Methods). A total of 89.9% and 80.7% of core and soft-core genes could be assigned to protein domains in the InterPro database—percentages nearly twice as high as those for shell and accession-specific genes (43.9% and 44.3%, respectively) (Extended Data Fig. 1e). The core and soft-core genes were expressed on average at a 2.2-fold higher level than the shell and accession-specific genes (Extended Data Fig. 1f), and showed markedly lower (1.7-fold on average) pairwise non-synonymous/synonymous substitution ratios ($K_a/K_s$) than did the shell genes (Extended Data Fig. 1g), suggestive of functional conservation. Functional enrichments of protein domains annotated in the InterPro database indicated that core and soft-core genes were enriched for domains that encode a wide range of functions involved in plant growth and development (Extended Data Fig. 1h,i), whereas domains related to retrotransposons and disease resistance were significantly enriched in shell and accession-specific genes (Extended Data Fig. 1h,i). These pan-genome resources provide a starting point from which to leverage the section-wide gene pool in potato biology and breeding.

## Phylogeny of *Petota* and neighbouring species

Owing to the lack of appropriate reference genomes, the evolutionary relationship among *Petota* and its sister sections *Lycopersicon* and *Etuberosum* is controversial[20,21]. Potato stolons are underground shoots or stems that are capable of bearing tubers[22], whereas *Etuberosum* species generate rhizomes resembling potato stolons, which grow upwards to form new daughter plants[23,24] (Supplementary Fig. 5). *Lycopersicon* species lack both rhizomes and stolons; thus, we hypothesized that *Etuberosum* is sister to *Petota* and *Lycopersicon* is the outgroup. To infer the evolutionary relationship among *Petota*, *Etuberosum* and *Lycopersicon*, we sequenced and de-novo-assembled two *Etuberosum* species—*Solanum etuberosum* and *Solanum palustre*—using PacBio continuous long reads; this resulted in 684.6-Mb and 738.9-Mb assemblies with contig N50 sizes of 3.9 Mb and 2.5 Mb, respectively. The completeness of these assemblies was estimated to be 95.6% and 95.6% by BUSCO (Supplementary Table 1).

By applying super-matrix and multispecies coalescent methods[25], we inferred a bifurcating species tree of 22 species from *Petota*, 2 from *Etuberosum* and 3 from *Lycopersicon*, as well as 2 outgroup species (*Solanum melongena* and *Solanum americanum*). The tree topologies were congruent, at major internal nodes, using both approaches (Fig. 1a, Extended Data Fig. 3a,b and Supplementary Table 3). We also estimated that *Etuberosum* diverged from the common ancestor of *Lycopersicon* and *Petota* at 8.30 million years ago (Ma; 95% highest posterior density interval: 7.9–8.8 Ma) (Supplementary Fig. 6). These results suggest that, with the genomic data that are available at present, *Etuberosum* is sister to the common ancestor of tomato and potato—in contrast to the hypothesis that *Etuberosum* is evolutionarily more closely related to *Petota* than *Lycopersicon*.

Phylogenetic topologies that are based on a single gene or genomic region may disagree with species topologies that are inferred from whole-genome markers[26]. We then split whole-genome alignments into 100-kb non-overlapping windows and applied phylogenetic inference for each window. This resulted in 1,899 trees with distinct topologies (Supplementary Tables 4 and 5), which suggests the widespread phylogenetic discordance of tree topology across the genome. Of these, 334 (17.6%) supported *Etuberosum* being a sister clade to *Petota* (Fig. 1a and Extended Data Fig. 3c). Given the recent divergence among *Petota*, *Etuberosum* and *Lycopersicon*, the lineage sorting processes might be incomplete among species in these sections. We observed 21.6–24.7% of the potato genome exhibiting incomplete lineage sorting (ILS) by comparing allele frequencies using a previously described method[27]. Interspecific hybridization has been prevalent among evolutionarily closely related species[28]. Using $D$ statistics[28], we detected gene flow between the species in *Petota* and *Etuberosum* sections ($D = 18.9\%$, $Z = 30.6$; Fig. 1c), and $f_4$-ratio statistics[28] showed that 8.4% of the potato genome showed admixture between *Petota* and *Etuberosum*. Similarly, we also observed the existence of ILS (Supplementary Fig. 7) and frequent gene flow (Extended Data Fig. 3d and Supplementary Fig. 8) among species within *Petota*, which was also reported in a previous study[5], and these may contribute to a lack of topological consensus of their evolutionary relationships (Fig. 1a). The pervasive inter- and intra-section phylogenetic discordance that we describe here suggests that potato evolution has a complex history that includes ILS and interspecific hybridization.

## Expansion of the repertoire of resistance genes

Clonal propagation gave rise to the emergence of tuber-borne diseases; potato has possibly evolved an expanded repertoire of resistance genes against these diseases[29], which might alter the genetic landscape of the potato immune system. Genes that encode nucleotide-binding domain and leucine-rich repeat (NLR) proteins have pivotal roles in plant immune signalling[30]. An accurate understanding of NLR evolution in potato species requires a comprehensive NLR dataset. However, plant NLRs occur mainly in genomic clusters, which makes their annotation challenging when using conventional approaches[31]. To mitigate this problem, we developed an 'NLR local annotation' pipeline and benchmarked it with a tomato NLR dataset, based on resistance gene enrichment sequencing (RenSeq), resulting in comparable numbers of NLRs (Methods and Supplementary Fig. 9). This resulted in 57,683 NLR genes, with the NLR copy number varying greatly among potato

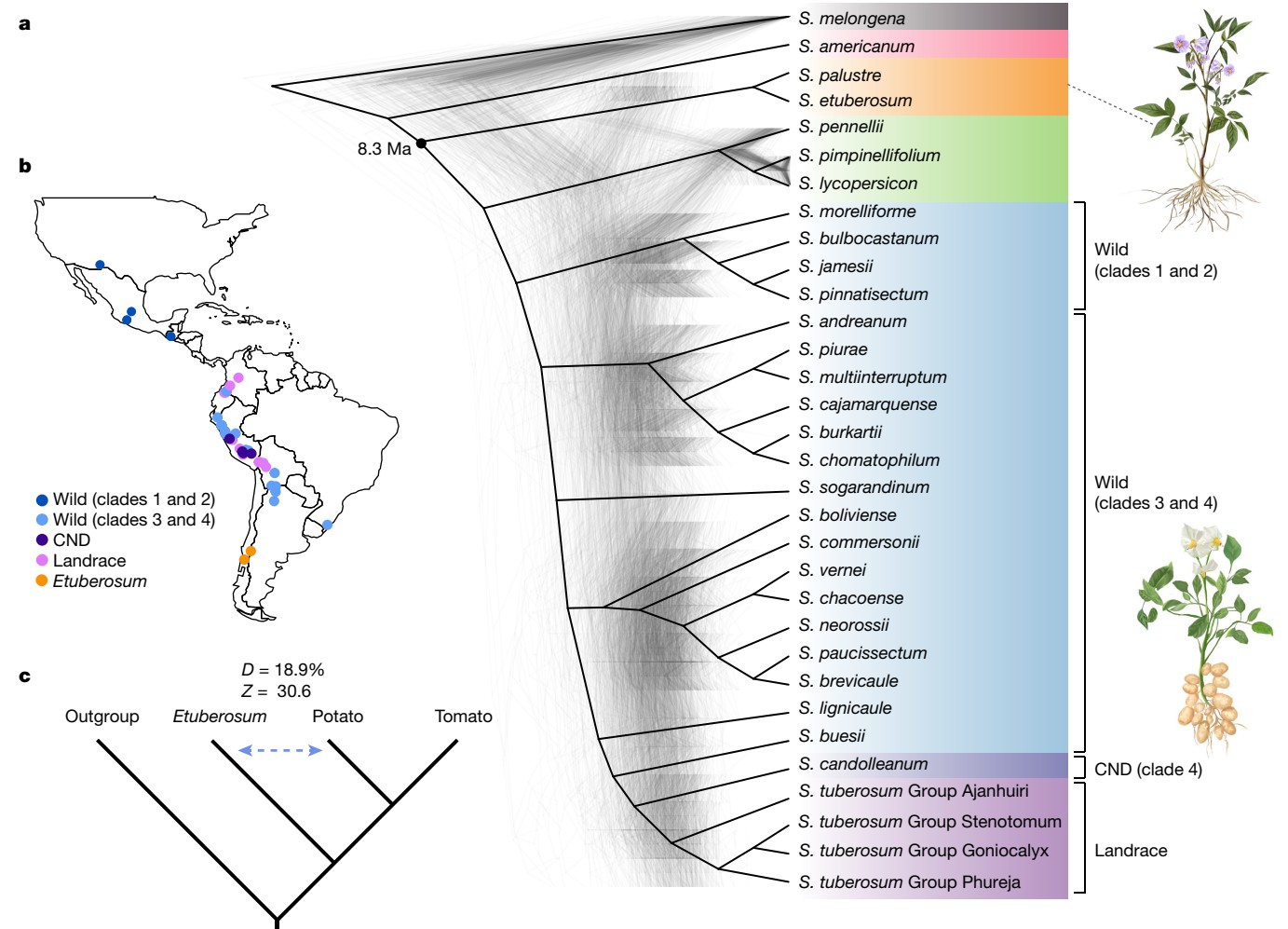

**Fig. 1 | Geographical distribution and phylogeny of the *Solanum* genus.**
**a**, Five hundred phylogenetic window trees (light grey lines) were randomly selected for visualization from 100-kb non-overlapping regions across the genome. The main cladogram shown here was built from 3,971 single-copy genes, based on 29 species (32 accessions, in which 4 are from *S. tuberosum*) (Supplementary Table 1). The number labelled beside the tree indicates the estimated divergence time. The pictures illustrate the morphological

differences of tuber-bearing and non-tuber-bearing species. **b**, Geographical origin of 39 samples (Supplementary Table 1) for which the longitude and latitude information are available. The base map was generated using the function mapBubbles() in the R package rworldmap. **c**, ABBA-BABA analysis of gene flow between *Petota* and *Etuberosum* species. Significant introgression events are detected between *Petota* and *Etuberosum*.

species—from 478 in *Solanum morelliforme* (PG1011) to 1,976 in *Solanum chacoense* (PG4042)—implying that immune systems in potato species have a diverse evolutionary history (Fig. 2a, Extended Data Fig. 4 and Supplementary Table 6).

We predicted 280–344 NLRs in the *Etuberosum* and tomato genomes and observed a significant expansion of NLRs in the potato MTG assemblies (Fig. 2b). We next classified NLRs from *Etuberosum*, tomato and potato, as well as 29 functionally validated NLRs, into 424 clusters, and identified 161 clusters that were putatively expanded in potato (Wilcoxon rank-sum test, $P < 0.05$; Supplementary Table 7). These clusters include some well-studied potato R gene families that confer resistance to the devastating late blight pathogen *Phytophthora infestans*, such as *R3a* (Fig. 2c and Supplementary Fig. 10). We identified 19,241 potato-specific NLRs that are present in potato, but absent in tomato and *Etuberosum*, and around 31.4% of them were expressed in stolon or tuber, suggesting that these may contribute to protecting stolons or tubers from pathogen infection (Supplementary Table 8).

Notably, we observed a similar NLR expansion event in wild relatives of the cultivated sweet potato, *Ipomoea trifida* and *Ipomoea triloba* (547 and 569 NLR genes, respectively), which are able to propagate

clonally[32], as compared with Japanese morning glory, *Ipomoea nil* (138 NLR genes) (Fig. 2b and Supplementary Table 9). *I. nil* did not have a vegetative reproduction organ[33], and diverged from *I. trifida* around 6.4 Ma (ref. [34]), a similar time point to that at which potato and tomato diverged (around 7.3 Ma)[35], which suggests that an expansion of the NLR repertoire might have co-evolved with the emergence of a vegetative mode of propagation.

## Tuber identity gene

The tuber, a storage and reproductive organ that confers a distinctive survival advantage to potato[36], has recently evolved throughout the divergence between tomato and potato. Despite some advances in our understanding of tuber development[8,37], insights into the evolution of tubers remain elusive. Previous studies have reported that potato recruited existing genes for new pathways that contributed to tuberization, suggestive of newly evolved *cis*-regulatory elements (CREs). Given the key role of conserved non-coding sequences (CNSs), which function as CREs in regulating gene expression and organogenesis[38,39], we identified 149,663 potato-specific CNSs (6.9 Mb) by computing conservation

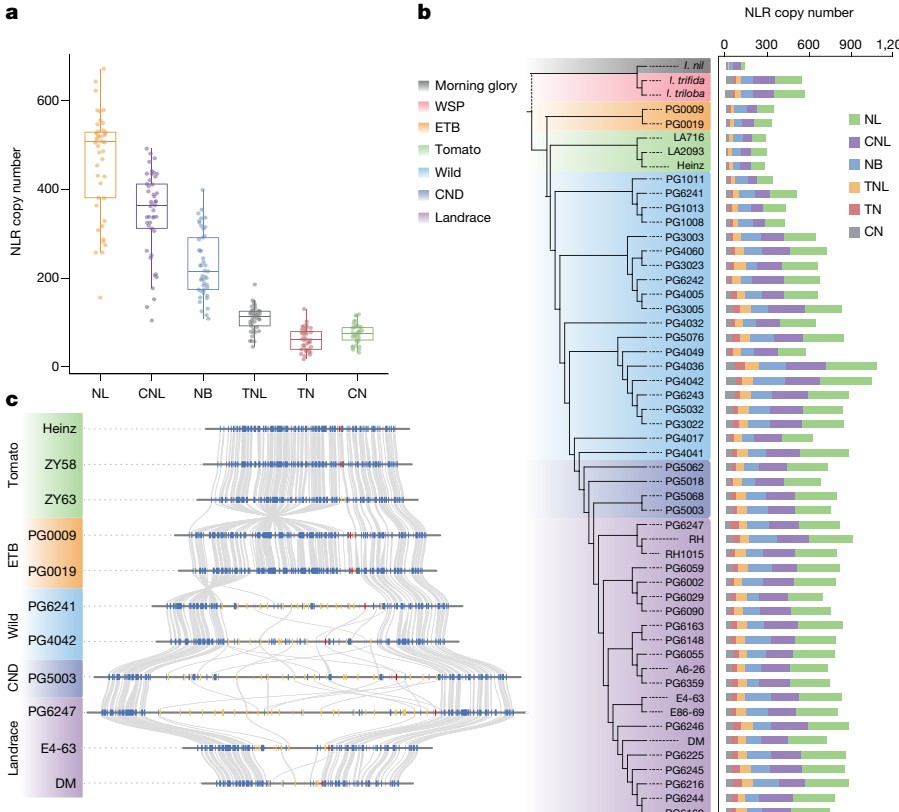

**Fig. 2 | Evolution of resistance genes in potato. a**, Canonical NLR copy number in potato. The upper and lower edges of the boxes represent the 75% and 25% quartiles, the central line denotes the median and the whiskers extend to 1.5 times the interquartile range (IQR). NL, NB-LRR; CNL, CC-NB-LRR; TNL, TIR-NB-LRR; TN, TIR-NB; CN, CC-NB; NB, NB domain only (see 'Reannotation and classification of nucleotide-binding resistance genes' in the Methods for detailed definitions of the abbreviations). Each NLR class contains 45 potato genomes. ETB, *Etuberosum*; WSP, wild relatives of sweet potato. **b**, Comparison of NLR copy numbers among different accessions. The NLRs from the potato monoploid assembled contigs were kept. The various colour backgrounds indicate accessions from different clades. **c**, Synteny plots of the *R3a* locus from 11 representative accessions. Red boxes indicate *R3a* orthologues in each accession. Yellow boxes indicate for NLR gene models and blue boxes denote other genes. Grey lines identify syntenic gene pairs.

scores, based on whole-genome alignment, using genome sequences of 45 potatoes (including DM), 24 tomatoes and 2 *Etuberosum* species. A total of 54.4% of these CNSs were localized at introns, followed by promoters (18.4%), intergenic regions (14.7%), downstream regions (5.0%), 3'-untranslated regions (3'UTRs; 4.4%) and 5'-UTRs (3.0%), which could potentially affect the expression of 17,871 genes (Extended Data Fig. 5a–c and Supplementary Table 10).

To identify candidate pivotal genes that are involved in tuber development, we identified 732 genes that are predominantly expressed in the stolon or tuber, among which 229 were associated with potato-specific CNSs and are also conserved among the 45 potato accessions (Fig. 3a and Supplementary Table 11). These genes encompassed 28 transcription factors, of which only one belongs to the plant-specific TCP transcription factor family (*Soltu.DM.06G025210*) (Extended Data Fig. 5d). Previous studies revealed that the TCP family is involved in the regulation of plant axillary meristem development; for example, rice tillering, maize branching and the development of cucumber tendrils[40,41]. The associated CNSs of this TCP were found in the −376 bp to −157 bp upstream from its start codon (Fig. 3b and Supplementary Fig. 11), suggesting putative regulatory roles in the TCP expression. Furthermore, transcriptomic data indicated that this gene was predominantly expressed in potato stolons (Fig. 3c), whereas the expression of its tomato orthologue (*Solyc06g069240.2*) could barely be detected. These findings suggest that recruitment and neofunctionalization of this gene may coincide with the emergence of tuber-bearing traits in the divergence of tomato and potato lineages.

To examine the function of *Soltu.DM.06G025210*, we generated knockout mutants by CRISPR–Cas9-based genome editing in the diploid *S. tuberosum* Group Phureja S15-65 clone (Extended Data Fig. 6a). The stolons of mutants were converted into branches, instead of swelling at the sub-apical region, during tuber initiation (Fig. 3d,e). Only under suitable growth conditions, and for a sufficient time, could the mutants generate a few small tubers (Extended Data Fig. 6b). These data suggest that *Soltu.DM.06G025210* is key to the initiation of potato tubers; we therefore named this gene *Identity of Tuber 1* (*IT1*).

Of note, a similar non-coding sequence (identity 94.6%) was identified upstream of *IT1* orthologues in the genomes of *Etuberosum* species (*IT1*^etb^), which are not capable of bearing tubers[23] (Supplementary Fig. 5). We then noted that *IT1*^etb^ was highly expressed in the rhizomes of *Etuberosum* species (Fig. 3c), which implies that *IT1* collaborates with additional genes in regulating tuber initiation, the functions of which may have been lost in *Etuberosum*.

We next performed yeast-two-hybrid library screening to identify putative IT1 interactors. Notably, SELF-PRUNING 6A (SP6A), the vascular-mobile signal in tuberization[42], was identified as interacting with IT1; this finding was further verified by firefly luciferase complementation imaging assays (Fig. 3f and Extended Data Fig. 7a), and suggests that SP6A and IT1 might act as a protein complex in regulating tuber initiation. We then analysed *SP6A* sequence variations in potato and *Etuberosum* genomes and found that the fourth exon in *SP6A*^etb^ was deleted, leading to an impaired phosphatidylethanolamine-binding protein (PEBP) domain (Fig. 3g and Extended Data Fig. 7b,c). Furthermore, quantitative PCR analyses did not detect any expression of *SP6A*

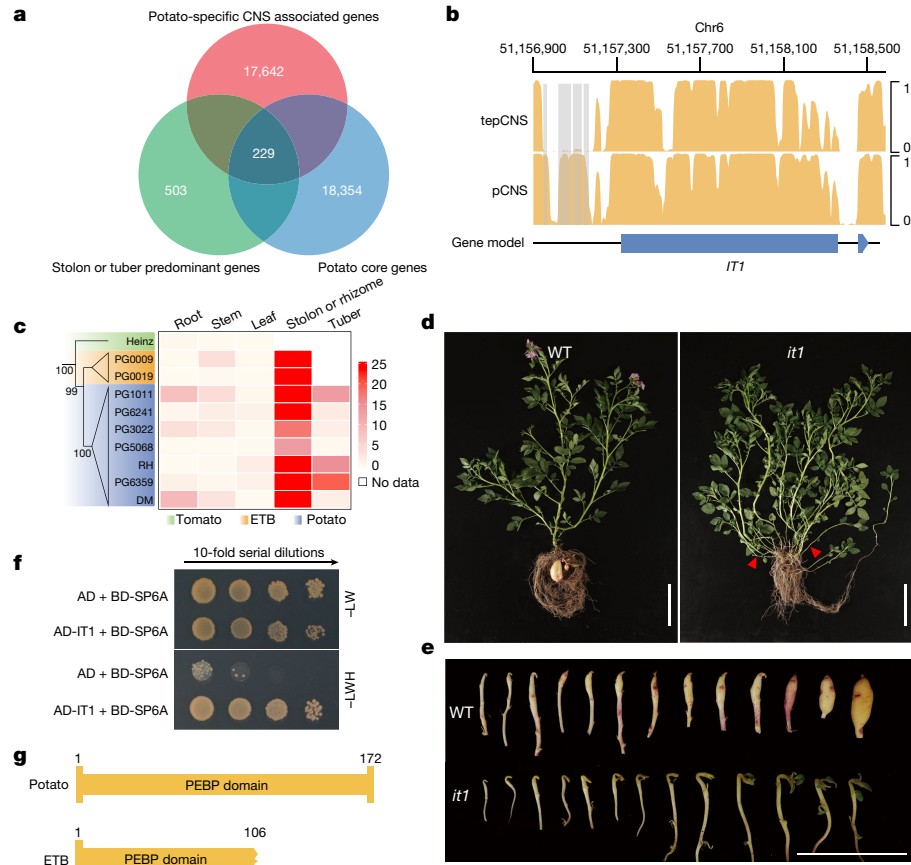

**Fig. 3 | Identification of a potato tuber identity gene. a**, Venn diagram describing the identification of 229 candidate genes that are involved in regulating stolon or tuber development. **b**, Conserved CNSs around the *Identity of Tuber1* (*IT1*) locus. tepCNS, conservative score for each site calculated from tomato, *Etuberosum* and potato genomes; pCNS, conservative score for each site calculated from 45 potato genomes. Grey blocks show potato-specific CNSs. **c**, Expression pattern of *IT1* and its orthologues in five tissues of *Etuberosum*, tomato and potato species. The 5-kb sequences up- and downstream of *IT1* from tomato, *Etuberosum* and potato were used to infer the phylogenetic relationships. **d**, Phenotypes of the *it1* knockout mutant. Red arrowheads indicate several abnormally developed stolons in the *it1* mutant. Scale bars, 10 cm. WT, wild type. **e**, Comparison of potato tuber development between wild type and the *it1* mutant. Scale bar, 5 cm. **f**, IT1 directly interacts with SP6A, as validated in a yeast-two-hybrid assay. Three independent biological experiments were performed. **g**, Domain architecture of SP6A in potato and *Etuberosum* species. AD, Gal 4 activation domain; BD, Gal4 DNA-binding domain; -LW, synthetic dropout medium without Leu and Trp; -LWH, synthetic dropout medium without Leu, Trp and His.

in *Etuberosum* leaves, under either long-day or short-day conditions (Extended Data Fig. 7d). These data suggest that the impaired function of *SP6A* may contribute to the non-tuber-bearing phenotype of *Etuberosum*. Further phylogenetic analysis, using 5 kb up- and downstream sequences of *IT1*, revealed that *Etuberosum*—rather than *Lycopersicon*—was sister to potato species (Fig. 3c), suggesting that *Etuberosum* represents a transitional form during the evolution of tuberization.

## Pan-genome-guided hybrid potato breeding

We previously developed the first generation of highly homozygous inbred potato lines using genome design, and the resultant hybrids showed strong heterosis[3,43]. For successful hybrid potato breeding, more inbred lines of high homozygosity are essential, and the first set of inbred lines also require further improvement, for which this pan-genome map can offer critical guidance.

To survey the level of homozygosity of the accessions studied—a key parameter for selecting starting materials for the development of inbred lines—we localized our alternate assembled contigs (ATGs; heterozygous genomic segments) and MTGs to the DM reference genome, and defined heterozygous and homozygous regions, respectively. We found that the length of heterozygous regions varied in the 41 accessions from 103 Mb (PG1011) to 710 Mb (PG5068) (excluding

inbred lines; Extended Data Fig. 8a and Supplementary Table 12). Within these heterozygous regions, we found 208–13,364 hemizygous genes in the 41 potato accessions, accounting for 0.5%–18.3% of predicted protein-coding genes, which was positively correlated to the estimated heterozygosity ($R^2 = 0.69$, $P = 1.85 \times 10^{-11}$; Supplementary Table 13). The distribution of homozygous genomic segments is a key indicator for the absence of large-effect deleterious mutations, which are less likely to be retained in the homozygous state because they are mostly recessive. In the case of tight linkage of two or more large-effect deleterious mutations in repulsion phase, some heterozygous segments will be retained in high-generation inbred lines[3,12]. The map of homozygous segments and gene hemizygosity, presented here, therefore offers potential targets to replace the corresponding heterozygous segments in the development of inbred lines to be used for diploid hybrid breeding.

To assess the genomic divergence in *Petota*, we performed whole-genome alignments, using tomato as a control. Just 204.4 Mb (28.0% of the DM reference genome) of genomic regions shared in potato landraces were identified, in contrast to the markedly higher 675.8 Mb (87.0% of the Heinz 1706 reference genome) in cultivated tomatoes (Supplementary Table 14). The average proportion of syntenic genes among potato landraces (61.3%) was also considerably lower than that among cultivated tomatoes (91.0%; Supplementary Table 15 and Extended Data

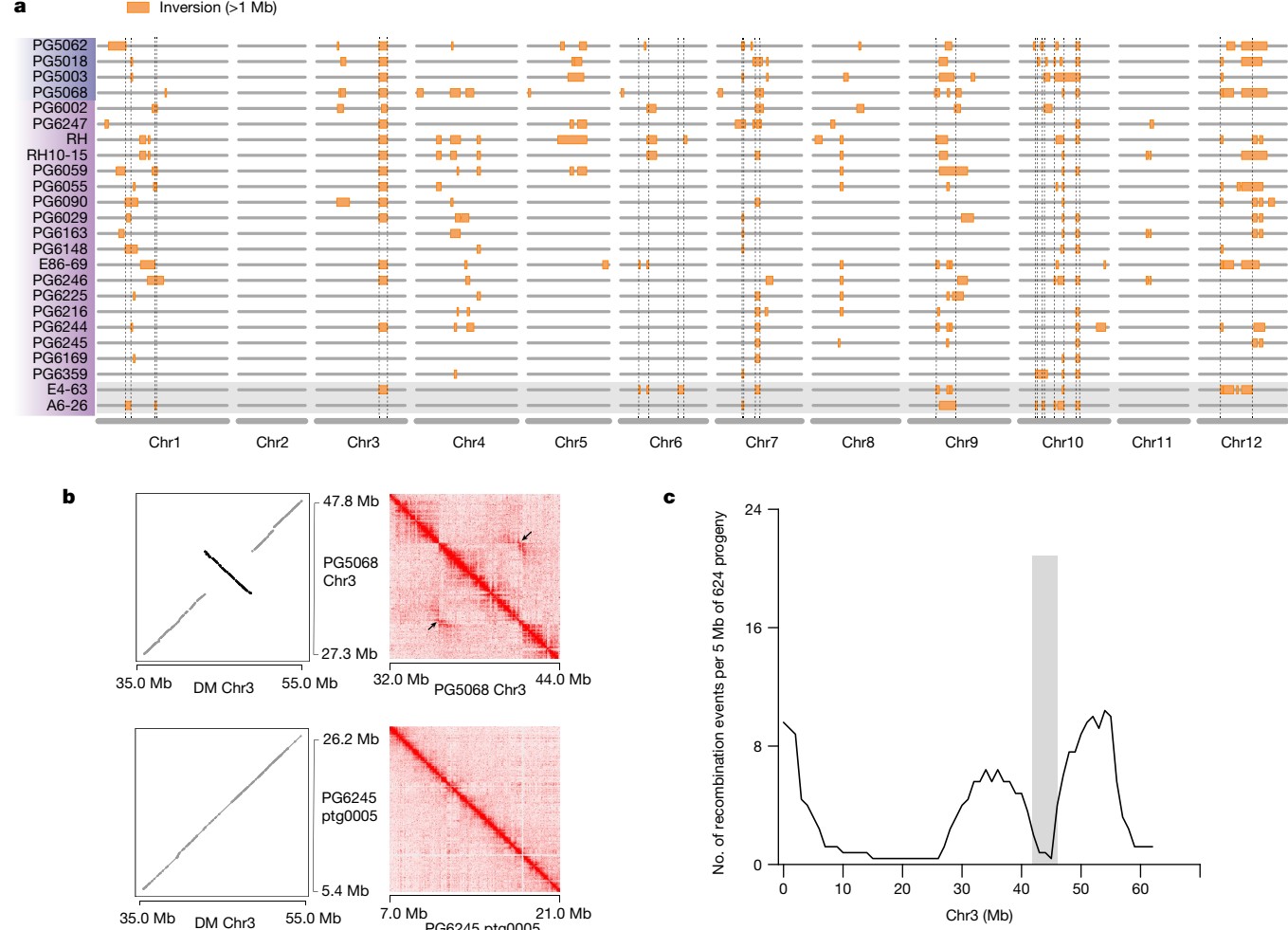

**Fig. 4 | Pan-genome-based map of large inversions. a**, Inversion map of 20 landraces and 4 CND accessions. The orange rectangles denote megabase-scale inversions. The dashed lines mark the regions containing inversions presented in either E4-63 or A6-26. **b**, The Hi-C-validated 5.8 Mb inversion event, using DM as the reference genome. Hi-C contact maps at 25-kb resolution for accession PG5068 (wild/CND haplotype) and PG6245 (DM haplotype), using Hi-C data from the homozygous line A6-26 (DM haplotype). Wild/CND haplotype, accessions carrying the inversion; DM haplotype, accessions without the inversion. **c**, Number of recombination events per 5 Mb on chromosome 3. The grey bar indicates the region with reduced recombination around the 5.8-Mb inversion.

Fig. 8b,c). This indicates a loss of synteny in cultivated potatoes, a critical genomic feature with implications for hybrid breeding.

The accumulation of structural variations (SVs) may contribute to the loss of synteny. We next identified 561,433 high-confidence SVs (more than 50 bp in size), affectting 167 Mb of the DM reference genome, of which 55.5% were rare (minor allele frequency < 0.05; Extended Data Fig. 8d). SVs close to genes might lead to the alteration of expression levels[44]. Most SVs (around 58.2% on average) were located in 5-kb upstream and downstream regions, followed by around 22.0% and around 13.4% overlapping intergenic and intron regions, whereas only around 6.4% affected exons (Extended Data Fig. 8e).

Among these, large inversions have been reported to suppress recombination by reducing crossing over[45]; this results in severe linkage drag when conducting backcross breeding, a tool that is required to improve the first-generation inbred lines. To avoid this problem, it is necessary to select donor lines without inverted fragments that contain target genes. Therefore, we constructed a map of large-scale inversions among the 20 landraces and the 4 *S. candolleanum* accessions, comprising 224 identified inversions with sizes ranging from 1.0 Mb to 17.6 Mb (Fig. 4a). Notably, an approximately 5.8-Mb paracentric inversion on the long arm of chromosome 3 (DM chr03: 42.9–48.7 Mb)—validated by examining chromatin interaction intensity (Fig. 4b)—co-segregates with the Y locus that controls carotenoid content in the tuber[3] (yellow flesh colour), in an $S_1$ population of 624 individuals (Supplementary Table 16). Our analyses of genetic mapping, association study and gene expression (Extended Data Fig. 9), together with previous studies of gene silencing[3,46,47], indicate that *Soltu.DM.03G018410* defines the Y locus. *Soltu.DM.03G018410* encodes a β-carotene hydroxylase (BCH) that controls the accumulation of zeaxanthin, which confers yellow colour in tuber flesh[47]. This gene was located around 1.5–2 kb proximal to the breakpoint of the 5.8-Mb inversion and is genetically inseparable with the inversion that contains 464 genes (Fig. 4c). Therefore, selection of individuals with yellow tuber flesh, a nutritional trait, may lead to severe linkage drag of unexpected phenotypes. With the aid of the constructed pan-genome-based inversion map, breeders could now select appropriate donor or acceptor lines for backcrossing.

## Discussion

The 44 high-quality genomes and the prevalent genetic variations identified herein offer useful resources for pan-genome-based potato breeding. These data are freely accessible through a comprehensive web-based Pan-Potato Database (http://solomics.agis.org.cn/potato/). The resources also enable the further construction of a pan-genome

reference integrating the genomes and variants of the 44 diverse potato accessions, which has the potential to minimize the effect of reference bias. The discovery of IT1 and its interaction with the mobile tuberization signal, SP6A, will pave the way for further elucidation of the evolution of tuber development. We also noticed that functional complementation of *SP6A* in *Etuberosum* may not induce tuber formation, according to a study on potato and *Etuberosum* heterografts[48], which suggests that the IT1–SP6A complex is necessary but not sufficient for tuberization.

Geographical isolation between the North and South American continent may contribute to the species from clades 1 and 2 being the sister lineage to wild species in clades 3 and 4 and other landraces. This is supported by our genomic data (Fig. 1a). Previous reports indicated that diploid cultivated potatoes were domesticated from wild potatoes from clade 4 (refs. [14,15]). In this study, we also found that *S. candolleanum* is sister to cultivated potato, further supporting this species as the immediate progenitor of cultivated potato. Further studies, coupled with the access to phased tetraploid potato assemblies, will allow the examination of introgression patterns from wild species, as introgression breeding was mainly conducted in these tetraploid cultivars. Considering that the endosperm balance number (EBN), a hypothetical unified prediction concept of crossability, between most of the wild species (17 out of 24) investigated here and the cultivated potatoes is the same (2EBN) (ref. [5]), the pan-genome will motivate attempts for the introgression of favourable traits from these wild species to breed inbred lines with better disease resistance and stress tolerance.

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

## Methods

### Sample selection and sequencing

We selected 44 representative potato accessions, 3 of which are publicly accessible[3,13], on the basis of phylogenetic relationships of 432 accessions (PRJNA378971, PRJNA394943 and PRJNA766763; genotype information is available at http://solomics.agis.org.cn/potato/ftp/Genotype_432sp/; Supplementary Fig. 1). To infer the phylogeny of the 432 accessions, reads were mapped to the DM v4 reference genome using BWA (0.7.5a-r405)[49], and single-nucleotide polymorphisms (SNPs) were then extracted using SAMtools (v.1.9)[50] and BCFtools (v.1.9)[49]. Fourfold degenerate SNPs with base quality ≥ 40 and mapping quality ≥ 30 were fed into IQ-TREE v.2.0.6 (ref. [51]), with parameters '-st DNA -m 012345 -B 1000'. In addition, two non-tuber-bearing species from the *Etuberosum* section PG0019 (*S. etuberosum*) and PG0009 (*S. palustre*) were chosen to be used in phylogeny inference (Supplementary Table 1). Sequencing of these 44 potato accessions was performed on the Pacific Biosciences Sequel II platform, in the circular consensus sequencing (CCS) mode, and the two *Etuberosum* species were sequenced on the Pacific Biosciences Sequel II platform, in the continuous long read (CLR) mode. A total of 15.9–38.1 Gb of HiFi reads was generated using CCS (https://github.com/PacificBiosciences/ccs) for the 41 newly sequenced potato accessions. For the construction of Hi-C libraries, DNA was extracted from in vitro seedlings, of which PG5068, PG0019 and E86-69 were digested with the restriction enzyme MboI, and PG6359 was digested with HindIII using the previously described Hi-C library preparation protocol[52]. These Hi-C libraries were sequenced on an Illumina HiSeq X Ten platform. The total RNA of 23 accessions (Supplementary Table 1) from the tissues of roots, stems, leaves, stolons, tubers and flowers was extracted for the library construction. These libraries were subsequently sequenced on the DNBSEQ-T7 system at Annoroad Gene Technology, which produced around 6 Gb data for each tissue in each sample.

### De novo genome assembly of 44 potato and 2 *Etuberosum* accessions

Genome heterozygosity was estimated using a $k$-mer-based approach by GenomeScope2.0 (ref. [53]). Genomes of the 44 HiFi sequenced accessions were assembled by hifiasm[54] (https://github.com/chhylp123/hifiasm), using default parameters. The initial output of hifiasm (v.0.13) yielded a pair of assemblies: (1) the primary assembly (in hifiasm named p_ctg), representing a mosaic haplotype without purging; and (2) the alternate assembly (in hifiasm named a_ctg), which represents the alternate haplotype absent from the primary one. To facilitate downstream analyses, including inter-genomic alignment and comparison of gene copy numbers, we generated 'monoploid' genome assemblies, accompanied by their heterozygous assembled fragments. Haplotigs from the primary assembled contigs, with haplotypes collapsed (p_ctg.*), were then excluded using the purge_dups (v.1.01) software (https://github.com/dfguan/purge_dups) to generate the heterozygous-region-purged assemblies, which were then termed as monoploid assembled contigs (MTGs), indicative of monoploid genomes. The raw alternate assemblies from hifiasm (a_ctg.*), in addition to the contigs that have been removed by purge_dups, were concatenated as the alternate assembled contigs (ATGs) to be the heterozygous genomic segments (Supplementary Fig. 2). The two *Etuberosum* genomes PG0019 and PG0009 were assembled using CANU v1.8 (ref. [55]), and then two rounds of Pilon v.1.23 (ref. [56]) were applied for genome polishing, using available resequencing data. Pseudo-chromosomes of the seven potato accessions (A6-26, E4-63, PG6359, E86-69, RH, RH10-15 and PG5068) and one *Etuberosum* accession (PG0019) were built with Hi-C reads, using the Juicer (v.1.5) (ref. [57]) and 3D-DNA (v.180922) (ref. [58]) pipeline, with parameters '-m haploid -I 15000 -r 0'. The assembly completeness in genic regions was evaluated using the solanales_odb10 database (for Solanaceae species) of BUSCO v.4.1.4 (ref. [18]), with default parameters.

### Identification and annotation of repetitive elements

Transposable elements (TEs) were identified by the Extensive De-Novo TE Annotator (EDTA)[59] v.1.9.4, and the non-redundant TE libraries for each accession were passed into RepeatMasker v.1.332 (http://www.repeatmasker.org) to mask potential genomic repeats together with simple repeats and satellites, by default parameters.

### Prediction of protein-coding genes

Three distinct strategies, comprising ab initio prediction, homology search and expression evidence, were combined to generate the predicted gene models. HISAT2 (v.2.0.1-beta) (ref. [60]) was used to perform splice alignment of RNA-sequencing (RNA-seq) reads to the assembled genomes, with '--dta' parameter. Potential transcripts were then assembled, using StringTie (v.1.3.3b) (ref. [61]) with parameter '--rf'. BRAKER2 v.2.1.6 (ref. [62]) was then run to use the transcript assemblies as hints to generate predicted gene models from AUGUSTUS (v.3.4.0) (https://github.com/Gaius-Augustus/Augustus) and to train the hidden Markov model (HMM) of GeneMark-ET (v.3.67_lic) (ref. [63]). The parameters set in BRAKER2 were '--nocleanup --softmasking'.

Non-redundant human-curated plant homologous protein sequences, downloaded from the UniProt Swiss-Prot database (https://www.uniprot.org/downloads), combined with published peptide sequences of tomato and potato[8,10,11,13,35], were used as homologous protein sequences. These and the assembled transcripts from StringTie (v.1.3.3b) were passed to MAKER2 (v.2.31.11) (ref. [64]). Putative gene structures were then inferred and subsequently used as the training set to generate the HMM in SNAP (v.2013-02-16) (https://github.com/KorfLab/SNAP). MAKER2 was then run again, combining previously generated SNAP HMM, GeneMark-ET HMM and AUGUSTUS tuned species settings, along with the predicted gene models produced from the first round of MAKER2, to synthesize the final gene annotations. The longest transcript of each predicated gene model was considered as the representative.

For gene functional annotation, InterProScan 5.34-73.0 (ref. [65]) was applied to predict potential protein domains, based on sequence signatures, with parameters '-cli -iprlookup -tsv -appl Pfam'.

### Analyses of the protein-coding-gene-based pan-genome

All-versus-all BLASTP (v.2.2.30+) (ref. [66]) results of 2,701,787 peptide sequences of protein-coding genes, annotated from 44 potato accessions and the DM v.6.1 reference genome[11], were input into OrthoFinder (v.2.5.2) (ref. [67]) for gene clustering, in which the MCL algorithm[19] was enabled by setting the inflation factor to 1.5, resulting in 51,401 non-redundant pan-gene clusters. We classified those clusters into 4 categories: core gene clusters that were conserved in all the 45 individuals; soft-core gene clusters, which were present in 42–44 samples in our collection; shell gene clusters, which were found in 2–41 accessions; and accession-specific gene clusters, which contained genes from only 1 sample. To facilitate these analyses, if genes from the DM reference were present in one cluster, this gene was selected as the representative; otherwise, the gene with the longest encoded protein was chosen.

Simulation of pan-genome size in terms of number of protein-coding genes was performed by PanGP (v.1.0.1) (ref. [68]) using the totally random algorithm, with a number of combinations, at each given number of genomes, of 500, and with the sample replication time set to 30.

Non-synonymous/synonymous substitution ratios ($K_a/K_s$) within core, soft-core and shell gene clusters were computed using ParaAT (v.2.0) (ref. [69]), with parameters '-m muscle -f axt -k'. The default parameter of KaKs_Calculator was set to estimate the $K_a/K_s$ values, which means that the $K_a/K_s$ value was the average of the output from 15 available algorithms comprising 7 original approximate methods (NG, method from Nei and Gojobori; LWL, method from Li, Wu and Luo; MLWL, modified method from Li, Wu and Luo; LPB, method from Li, Pamilo and Bianchi; MLPB, modified method from Li, Pamilo and Bianchi; YN, method from

Yang and Nielsen; MYN, modified method from Yang and Nielsen), 7 gamma-series methods (γ-NG, γ-LWL, γ-MLWL, γ-LPB, γ-MLPB, γ-YN and γ-MYN) and one maximum likelihood method (GY, method from Goldman and Yang) (ref. [70]). To simplify the calculation, we randomly selected 1,500 clusters from clusters of core, soft-core and shell categories. Within each cluster, $K_a/K_s$ values between gene pairs from 50 randomly chosen combinations of 2 accessions were estimated. The non-parametric multiple comparisons Kruskal–Wallis test was used to perform significance analyses for sample median, using the agricolae package in R v.4.0.3 (https://www.r-project.org/), as these data did not comply with a normal distribution. Multiple comparisons were performed, using the Fisher's least significant difference. The level of significance used in the post-hoc test was 0.001. Functional enrichment was performed, using Fisher's exact tests in R. Those functional classes with $P < 0.05$ were regarded as significantly enriched.

## Whole-genome alignment of 73 *Solanum* species

Whole-genome alignment of 73 accessions, comprising 44 potato MTGs and the genomes of DM, 2 *Etuberosum* species, 24 tomato accessions (https://solgenomics.net/projects/tomato100, http://caastomato.biocloud.net/page/download/), and 2 outgroup species of *S. americanum* and *S. melongena* (http://eggplant-hq.cn/Eggplant/home/index)[35,44,71,72] were performed by ProgressCactus (v.1.2.3) (ref. [73]). The tomato genomes investigated in this study were all built using the third-generation sequencing technique (PacBio CLR and Nanopore) and are all assembled into 12 chromosomes, indicative of their relatively high qualities. The guide tree used in ProgressCactus was inferred by IQ-TREE, v.2.0.6 (ref. [51]). To reduce the computation requirement, genome sequences were soft-masked and contigs shorter than 100 kb were discarded. To facilitate downstream analyses, we next used PHAST toolkit v.1.5 (ref. [74]) to generate 73-way multi-alignment blocks in fasta format, relative to the DM genome.

## Genome comparison of 44 HiFi-assembled potato accessions

The 44 MTGs were aligned to the DM reference genome, using the nucmer program in MUMmer v.4.00rc1 (ref. [75]) software with the '--mum' parameter, and alignments with an identity of less than 90% and length shorter than 1,000 bp were discarded. We used a modified version of dotPlotly (https://github.com/tpoorten/dotPlotly/blob/master/mummerCoordsDotPlotly.R) for visualization. To assess the heterozygosity distribution of 41 diploids (excluding 3 homozygous inbred lines), their MTGs and ATGs were split into 5-kb fragments and were aligned to the DM reference genome, using the same approach described above, and alignments shorter than 5 kb were discarded to reduce potential noise.

## Identification of syntenic genes

To identify syntenic gene pairs, BLASTP (v.2.2.30+) was used to calculate pairwise similarities (*e*-value $< 1 \times 10^{-5}$), and MCscanX[76] with default parameters was then applied.

## Phylogenetic analyses

To build a super-matrix tree of 29 species (32 accessions, in which 4 are from *S. tuberosum*), amino-acid sequences of the longest transcripts of their annotated gene models were first extracted from the MTG genomes of 25 potatoes, 3 tomato accessions (see Supplementary Table 1 for more details)[35,44,71,72], 2 *Etuberosum* species, *S. americanum* and eggplant[77]. All-versus-all alignments were generated using DIAMOND (v.2.0.6.144) (ref. [78]). The results were then passed to OrthoFinder (v.2.5.2)[67] to infer orthology. A total of 3,971 single-copy orthologues gene clusters were then generated and 32-way protein alignments for these genes were computed using MAFFT (v.7.471) (ref. [79]) with default parameters. Maximum likelihood inference of phylogenetic relationships was performed using IQ-TREE v.2.0.6 (ref. [51]), by automatically

calculating the best-fit amino-acid substitution model via the '-m MFP' parameter. The consensus tree was generated specifying the number of bootstrap replicates as 1,000 by ultrafast bootstrap approximation[80]. We also constructed a phylogenetic tree using an additional 20 potato (including DM) and 21 tomato genomes by applying the same approach described above.

To minimize the effect of ILS, we applied a multi-species coalescent-based method incorporated in ASTRAL (v.5.7.8) (ref. [81]) to generate a species tree. ASTRAL took 3,971 single-copy gene trees as input and generated a species tree estimated by searching for the species tree that was most congruent with quartets garnered from the input gene trees.

To infer the local phylogeny among the 32 representative accessions, considering the diverse nucleotide evolution rate of coding and non-coding regions, we masked coding regions according to the gene prediction in DM using the maskFastaFromBed command embedded in BEDTools (v.2.29.2) (ref. [82]), and repetitive regions were then hard-masked. We split Cactus whole-genome alignment blocks into 100-kb non-overlapping windows and inferred tree topologies for each window, using IQ-TREE[51] with the parameter '-m GTR'. Next, we filtered the window trees with three standards: (1) fully aligned length > 10 kb; (2) missing rate < 20%; (3) mean bootstrap values > 80. After filtering, we next re-estimated the tree topologies of the retained 1,899 windows, using the selected best substitution model for each window, using ModelFinder implemented into IQ-TREE (ref. [51]). To help with visualization, 500 window trees were randomly selected, with an *R* script modified from a previous report[83] (https://zenodo.org/record/3401692#.YNrvJ6e76XQ). The consensus tree topology was generated by IQ-TREE[51], using concatenated single-copy protein-coding sequences identified by OrthoFinder[67].

## Estimation of the divergence time

BASEML and MCMCTREE in the PAML package (v.4.9) (ref. [84]) were used to estimate the divergence time. To reduce the computational burden, coding sequences (CDSs) of single-copy genes from 10 representative species (*S. melongena*, *S. americanum*, PG0019, LA716, LA2093, Heinz 1706, PG6241, PG4042, PG5068 and DM) were selected for a rough estimation of the substitution rate using BASEML with model = 7. MCMCTREE was then applied to estimate the divergence time with parameters 'model = 7, burnin = 500,000, sampfreq = 100, nsample = 20,000'. The divergence time of potato–tomato (7.3–8.0 Ma)[35,85] and potato–eggplant (13.7–15.5 Ma)[85–87] was used for calibration. The estimation was performed for two rounds and generated very similar results.

## ABBA-BABA statistics

On the basis of the genome assemblies, around 20-fold short reads of the 25 representative *Petota* accessions, 2 *Etuberosum* species, 3 tomatoes and *S. americanum* were simulated using WGSc (https://github.com/YaoZhou89/WGSc), and reads were mapped to the outgroup reference genome *S. melongena* using BWA-mem[49] with the default parameters. Bi-allelic SNPs were then identified using SAMtools[50] and BCFtools[49]. Setting *S. melongena* as the outgroup, an ABBA-BABA test was performed between all possible triplets among potato, tomato and *Etuberosum* species, using the Dtrios program within Dsuite (v.0.4 r28)[88], with the '-c' parameter. The tree topology among these four species, inferred from the whole-genome data in Newick format, was also passed into Dtrios via the '-t' parameter.

## Inference of ILS

The level of ILS at a given bi-allelic SNP *i* from the above mentioned 32-way alignment was calculated as $C_{ABBA(i)}$ and $C_{BABA(i)}$ divided by the total count of segregating sites: $(C_{BAAA(i)} + C_{ABAA(i)} + C_{AABA(i)} + 2(C_{BBAA(i)} + C_{BABA(i)} + C_{ABBA(i)}))/3$, as described previously[27]. The tree topology used was (((*Lycopersicon*, *Petota*), *Etuberosum*), *S. melongena*).

To evaluate the theta value for internal branch, which reflects the level of effective population size[89], we divided the mutation units by

coalescent units. The mutation units were inferred by IQ-TREE (ref. [51]) and the coalescent units were inferred by ASTRAL.

## ILS simulation

Simulation of 20,000 gene trees with ILS among six potato accessions (*S. tuberosum* Group Stenotomum, *S. candolleanum*, *Solanum lignicaule*, *S. chacoense*, *Solanum cajamarquense* and *Solanum bulbocastanum*) were performed by DendroPy (ref. [90]). The branch lengths of the estimated species tree by ASTRAL were used as an input. Frequencies between the observed and simulated gene-tree topologies from all possible four-species groups among the six potato species were plotted. The correlations were computed using the correlation function 'cor()' in R using the 'pearson' method.

## Identification of SVs

Both read-mapping-based and assembly-based approaches were applied to identify SVs (≥50 bp in length). SVIM (v.1.4.2) (ref. [91]) was used to identify putative SVs, consisting of insertions, deletions, inversions, duplications and translocations. SVs with quality ≥ 10 and number of supported reads ≥ 5 were kept. Assembled genomes of each accession were first aligned to the DM v.6.1 reference using the nucmer program embedded in MUMmer v4.00rc1 (ref. [75]), with the following parameters: '--batch 1 -c 500 -b 500 -l 100'. The alignments in delta format were passed to the delta-filter program to retain highly reliable alignments with length ≥ 100 bp and identity ≥ 90%. Assemblytics (v.1.2.1) (ref. [92]) was subsequently applied to identity SVs from the filtered alignments, setting the minimum SV size to 50 bp. To make the false positive rate in our SV dataset as low as possible, we only kept SVs in terms of insertions, deletions, inversions, duplications and translocations < 10 kb in size, identified by SVIM. For SV ≥ 10 kb, only insertions and deletions reported in Assemblytics were retained. The two SV datasets for each sample were then combined, using SURVIVOR (v.1.0.7)[93] merge with parameters '0 1 1 1 0 50'.

To detect megabase-scale inversion events among the 20 landraces and 4 CND accessions, we applied ragtag (v.2.1.0) (ref. [94]) with the default parameters, to order and orient the contig-level assemblies into 12 chromosomes, using the DM genome as the reference. Inversions were next identified using SyRI (v.1.4) (ref. [95]) with parameters '-k -F S'. Only those inversions that located in a single contig were retained for downstream analyses.

## Identification of hemizygous genes

To identify regions present in MTGs but absent in ATGs, we mapped HiFi reads of each accession to its corresponding MTGs using minimap2 (v.2.21-r1071) (ref. [96]), and heterozygous deletions were detected using SVIM (v.1.4.2) (ref. [91]) with default parameters (length ≥ 50 bp, quality ≥ 10, number of supported reads ≥ 2). To identify sequences present in ATGs but absent in MTGs, we aligned ATGs to MTGs from each accession and extracted the inserted regions using Assemblytics[92] with parameters 'unique_anchor_length = 10,000, min_variant_size = 50, max_variant_size = 10,000,000'. These results were merged as heterozygous SVs, and genes overlapping with those SVs were considered as hemizygous genes, as applied in a previous report[97].

## Analyses of recombination events

Breakpoints of crossing-overs were inferred based on the 624 selfing progenies of PG6359 (ref. [3]), using a method described previously[7].

## Reannotation and classification of nucleotide-binding resistance genes

NLR-annotator (v.0.7) (ref. [98]) was first applied to identify genomic segments containing putative nucleotide-binding domain and leucine-rich repeat (NLR) genes for each accession. A total of 7,007 amino-acid sequences of high-confidence NLR gene models, downloaded from resistance gene enrichment sequencing (RenSeq)-based NLR genes of 15 tomato accessions[99], putative NLR genes in *Arabidopsis*[100] (annotation version Araprot11) and experimentally validated NLR genes obtained from PRGdb 3.0 (ref. [101]) and RefPlantNLR[102], were used as homologous protein evidence. Training sets from SNAP and AUGUSTUS for each accession were then inputted to MAKER2, together with the assembled transcripts, in GFF3 format, and the homologous proteins, to predict and synthesize the final gene models. The reannotated NLR gene models were then integrated with the whole-genome annotation results, and originally predicted genes overlapping with our reannotated NLRs were removed to avoid redundancy.

To examine the completeness of NLR loci generated by our pipeline, we produced three NLR datasets of tomato accession 'Heinz 1706' from the predicted high-confidence and representative gene models (annotation version ITAG 4.0), predicted models using our pipeline, and previously reported RenSeq-derived models[99], respectively. The RGAugury (v.2.2) (ref. [103]) pipeline was then used to classify putative nucleotide-binding site (NB-ARC) domain-encoding genes into different subgroups, on the basis of domain and motif structures: TN (Toll/interleukin-1 receptor (TIR) and NB-ARC), CN (coiled-coil (CC) and NB-ARC), NL (NB-ARC and leucine rich repeat (LRR)), CNL (CC, NB-ARC and LRR), NB (NB-ARC), TNL (TIR, NB-ARC and LRR).

For identification of putatively expanded NLR clusters in potato, the annotated NLR loci from 45 potato, 22 tomato and 2 *Etuberosum* genomes were classified into clusters, using the method described in the pan-genome analysis. The NLR copy numbers in potato, tomato and *Etuberosum* accessions, in each cluster, were compared by Wilcoxon rank-sum test using the R package exactRankTests. The clusters with $P < 0.05$ were extracted as the expanded clusters. For the potato-specific NLR analyses, the NLR protein sequences from 2 *Etuberosum* species and 22 tomato species were merged together, as a query, to blast against those from the 45 potato species. If the best hit of a potato NLR showed an identity < 75, the NLR was defined as potato-specific. NLRs with transcripts per kilobase of exon model per million mapped reads (TPM) ≥ 1 in potato stolon or tuber were extracted and considered as expressed in these tissues.

## Gene expression profiling

RNA-seq reads of 23 accessions (Supplementary Table 1) as well as DM (SRA030516) were mapped to the corresponding assembled genome, using HISAT2 (v.2.0.1-beta) (ref. [60]), with parameters '-x --dta'. StringTie (v.1.3.3b) (ref. [61]) was applied to compute the expression level for each predicted gene in terms of TPM values, using '-e -G' parameters.

## Analyses of CNSs

Tools embedded in the PHAST package (v.1.5) (ref. [74]) were used for CNSs analyses. The msa_view program was applied to extract fourfold degenerate synonymous sites and to prepare sufficient statistics, on the basis of multiple alignments and CDS annotation of the reference genome. PhyloFit was then used to train the un-conserved model, with sufficient statistics generated by msa_view. phastCons, with the parameter '--most-conserved' used to identify conserved regions and assign an odds score for each site. Finally, conserved regions containing gaps and overlapping with CDS were removed to generate CNSs shared among potato species. To further remove CNSs shared within outgroup species, we identified CNSs from genomes of 45 potato, 24 tomato and 2 *Etuberosum* species, using the same pipeline presented above. The potato conserved CNSs that shared sequences with tomato and *Etuberosum* species were removed. In addition, short sequences (<5 bp) were excluded and sequences that were located within 10 bp of each other were merged to generate the final potato-specific CNSs. ChIPseeker v.1.24.0 (ref. [104]) was adopted to annotate these CNSs, in which sequences 3 kb upstream from the start codon or 3 kb downstream from the stop codon of a certain gene were defined as promoter or downstream regions. Genes possessing CNSs within their promoters, introns, upstream regions, downstream regions or UTRs were

defined as CNS-associated genes. pyGenomeTracks v.3.6 was applied to visualize the CNS region[105]. Sequences flanking *IT1* CNS regions were extracted from the 71-way multiple alignment and were imported into MView (v.1.67) (ref. [106]) to generate the multiple comparisons figure.

### Generation of *it1* mutants

The diploid *S. tuberosum* Group Phureja S15-65 clone was used for gene editing in this study. The 19-nucleotide single-guide RNA sequence for *IT1* from the S15-65 clone was incorporated into the CRISPR–Cas9 vector pKSE401 (ref. [107]). Three-week-old plantlets were used for transformation. *Agrobacterium*-mediated transformation of potato internodes was conducted as previously described[6]: after two days of pre-culture, the explants were co-cultured with *Agrobacterium* containing pKSE401 with the target sequence for two days, in the presence of 2 mg l$^{-1}$ α-naphthaleneacetic acid and 1 mg l$^{-1}$ zeatin, followed by callus induction and regeneration mediated by 0.01 mg l$^{-1}$ α-naphthaleneacetic acid and 2 mg l$^{-1}$ zeatin until shoot proliferation. Positive transformants were screened on the basis of growth on the medium containing 50 mg l$^{-1}$ kanamycin.

### Yeast-two-hybrid library screening

The cDNA of S15-65 stolons was used to construct the cDNA library for yeast-two hybrid by using the CloneMiner II cDNA Library Construction Kit. The library was screened, with the *IT1* as bait, according to the Matchmaker Gold Yeast Two-Hybrid System User Manual. To further validate the interaction between IT1 and SP6A, the CDSs of *IT1* and *SP6A* were inserted into pGADT7 and pGBKT7 vectors, respectively, and co-transfected into the yeast strain AH109, and the yeast cells were then plated on SD/−Leu/−Trp medium and SD/−Leu/−Trp/−His medium and cultivated at 30 °C for 5 days.

### Firefly luciferase complementation imaging assay

The CDSs of *SP6A* and *IT1* were fused in the pCAMBIA-nLUC-GW and pCAMBIA-cLUC-GW vectors, respectively[108]. The vectors were transformed into *Agrobacterium* strain GV3101, and infiltrated into *Nicotiana benthamiana* leaves. After 3 days, the detached leaves were sprayed with 100 mM luciferin and kept in the dark for 10 min. The leaves were observed under a low-light cooled charge-coupled device imaging apparatus, Lumazone 1300B (Roper Bioscience).

### Quantitative PCR analysis of *SP6A* expression

The seeds of potato inbred line E4-63 and *Etuberosum* species PG0019 were planted in soil and cultivated under long-day conditions (16-h light, 8-h dark) for one month, and then half of the plants were transferred to short-day (8-h light, 16-h dark) conditions. When flower buds developed in the long-day plants (usually two months after sowing), the fourth leaf of both long-day and short-day plants was harvested at ZT4 to investigate *SP6A* gene expression. The total leaf RNA was extracted using an RNAprep Pure Plant Kit (DP441), and a PrimeScript RT Reagent Kit (RR047A) was used to reversely transcribe the RNA to cDNA. Quantitative PCR was performed using SYBR Premix Ex Taq II (RR820A) on a 7500 Fast Real-Time PCR system (Applied Biosystems), according to standard instructions. *EF1A* was used as the internal reference. The specific primers used are listed in Supplementary Table 17.

### Syntenic analyses of *R3a* and *SP6A* loci

To plot syntenic relationships around the *R3a* locus, collinear blocks between the given two species were identified by MCScanX (Python version)[109]. Syntenic genes around *R3a* loci were extracted and plotted using in-house R scripts. For a synteny plot of the *SP6A* loci, the *SP6A* genomic sequences from different species were extracted and aligned using MAFFT, with '--auto' parameter[79]. In-house Python scripts were used to transfer aligned regions between two species to the BED format required by MCScanX. The micro-synteny plot between the two species

was then generated. Finally, synteny plots among different species were merged using Adobe Illustrator.

### Reporting summary

Further information on research design is available in the Nature Research Reporting Summary linked to this paper.

## Data availability

All PacBio sequence data, transcriptome data and Hi-C data in this study have been deposited at the National Center for Biotechnology Information (NCBI) Sequence Read Archive (SRA), with BioProject accession number PRJNA754534, and the National Genomics Data Center (NGDC) Genome Sequence Archive (GSA) (https://ngdc.cncb.ac.cn/gsa/), with BioProject accession number PRJCA006011. Genome assemblies, annotation, sequence variation and gene expression data for the 46 accessions and the genotype information of 432 lines that were used for sample selection are hosted in the Pan-Potato Database (http://solomics.agis.org.cn/potato/, http://218.17.88.60/potato/). Publicly available sequencing data were downloaded from the NCBI with BioProject accession numbers PRJNA641265, PRJNA573826, PRJNA378971, PRJNA394943 and PRJNA766763.

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

**Acknowledgements** We thank N. Stein, Y. Zhang, S. Dong, L. Wang, S. Zhou, L. Liu, Y. Wu and Y. Yang for discussion and comments on the project, and the AGIS CAAS-YNNU Joint Academy of Potato Sciences for greenhouse assistance. This work was supported by the National Key Research and Development Program of China (grant 2019YFA0906200), the Agricultural Science and Technology Innovation Program (CAAS-ZDRW202101), the Shenzhen Science and Technology Program (grant KQTD2016113010482651), the Special Funds for Science Technology Innovation and Industrial Development of Shenzhen Dapeng New District (grant RC201901-05), the National Natural Science Foundation of China (31902027), Shenzhen Outstanding Talents Training Fund and the China National Key Research and Development Program to C.Z. (2019YFE0120500).

**Author contributions** S.H. conceived and designed the project. D.T., Y.J., H.L. and S.H. wrote the manuscript. D.T., Y.J. and H.L. performed the bioinformatics analyses. L.C. and Z.B. assisted in bioinformatics analyses. J.Z., Z.L., S.F. and X.Z. performed the molecular work on *IT1*. P.W. and D.L. contributed to the greenhouse work. G.Z. provided the computing platform. H.W., Yao Zhou, Yongfeng Zhou, G.J.B., C.R.B. and C.Z. coordinated the project.

**Competing interests** The authors declare no competing interests.

**Additional information**
**Correspondence and requests for materials** should be addressed to Sanwen Huang.

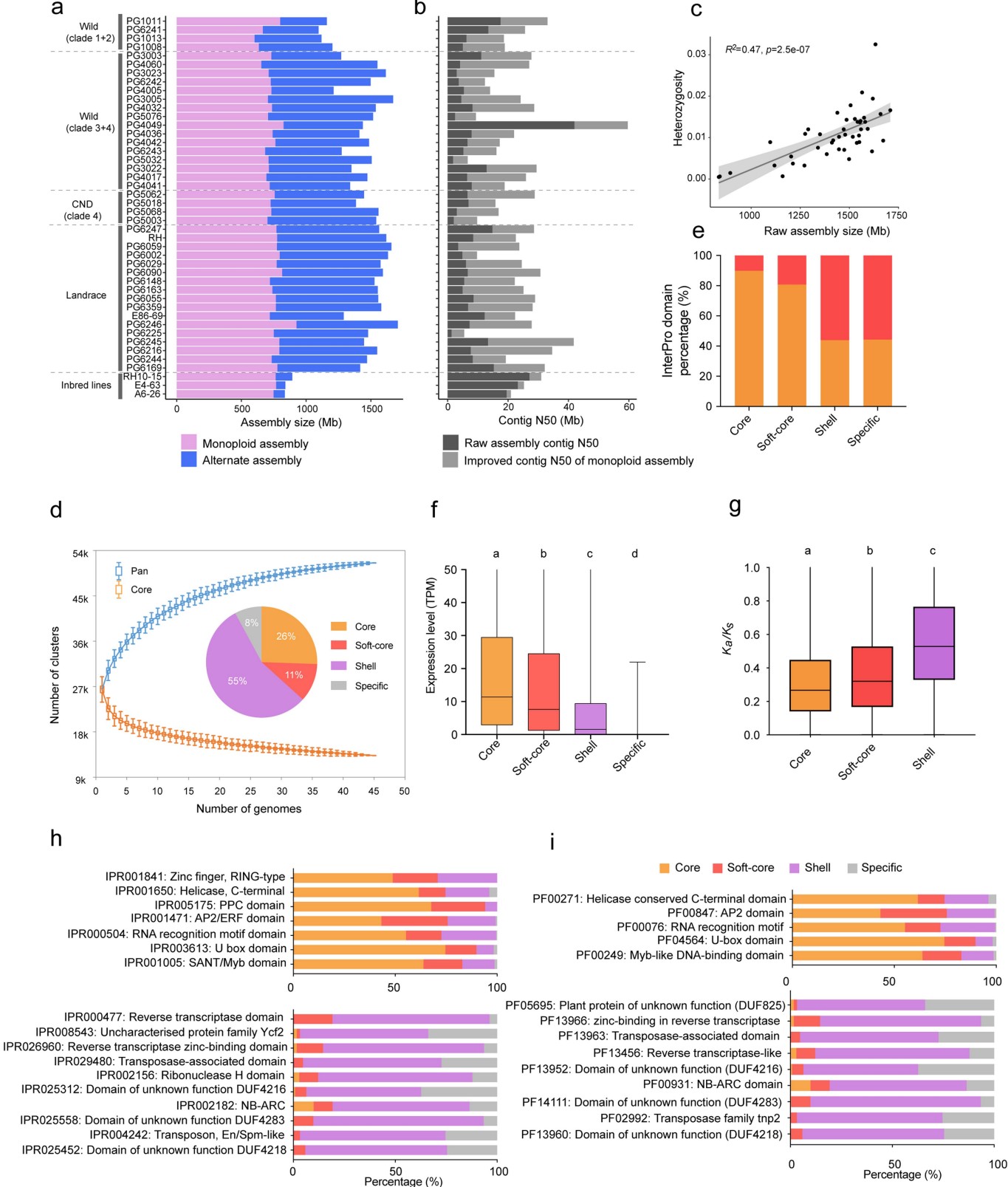

**Extended Data Fig. 1** | See next page for caption.

**Extended Data Fig. 1 | Pan-genome of 45 potato accessions. a**, Assembled size of monoploid assembled contigs (MTGs) and alternate assembled contigs (ATGs). **b**, Contig N50 of raw assembled contigs and improved contig N50 of MTGs. **c**, Correlation between raw assembly size and heterozygosity. The grey shaded region indicates the 95% confidence interval using a linear model ('lm'). **d**, Simulation of pan- and core-genome sizes, in terms of number of gene clusters and pan-genome composition. At each given number of genomes, the number of combinations is 500 with 30 times of replication. **e**, Percentage of genes in core, soft-core, shell and accession-specific gene subsets with annotated InterPro protein domains. Orange bars show the proportion of genes with InterPro domains, whereas red bars depict the genes without those domains. **f**, Expression profiles of genes belonging to core (13,123), soft-core (5,732), shell (5,009) and accession-specific (134) gene families. **g**, Non-synonymous/synonymous substitution ratios ($K_a/K_s$) within core, soft-core, and shell genes. Kruskal-Wallis test was used to determine significance. Multiple comparisons were performed, using the Fisher's least significant difference. The level of significance used in the *post hoc* test was 0.001. Number of gene pairs used in core, soft-core and shell genes are 52,148, 28,363 and 31,654, respectively. The upper and lower edges of the boxes represent the 75% and 25% quartiles, the central line denotes the median and the whiskers extend to 1.5 × IQR in **d**, **f** and **g**. **h**, InterPro protein domain enrichments of core and soft-core (upper panel) and shell and accession-specific (lower panel) genes relative to pan genes. **i**, Pfam protein families enriched in core and soft-core (upper panel) and shell and accession-specific (lower panel) genes, relative to pan genes.

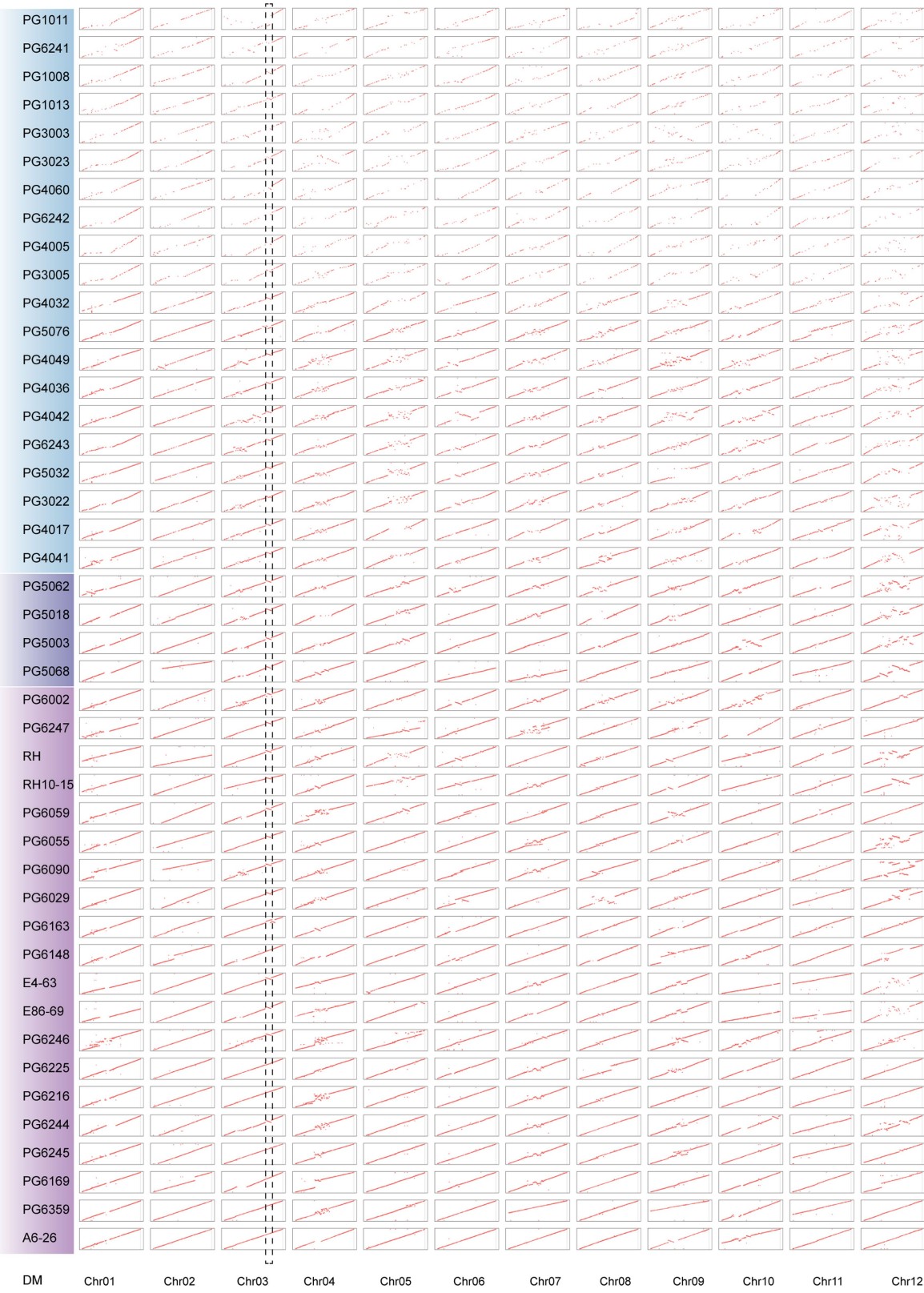

**Extended Data Fig. 2 | Genome-wide alignments among 45 genome accessions.** Whole-genome alignments of 44 MTGs to DM reference genome. Alignments with length greater than 10 kb and showing greater than 90% identity are kept for visualization. Black dashed rectangles indicate the specially focused regions.

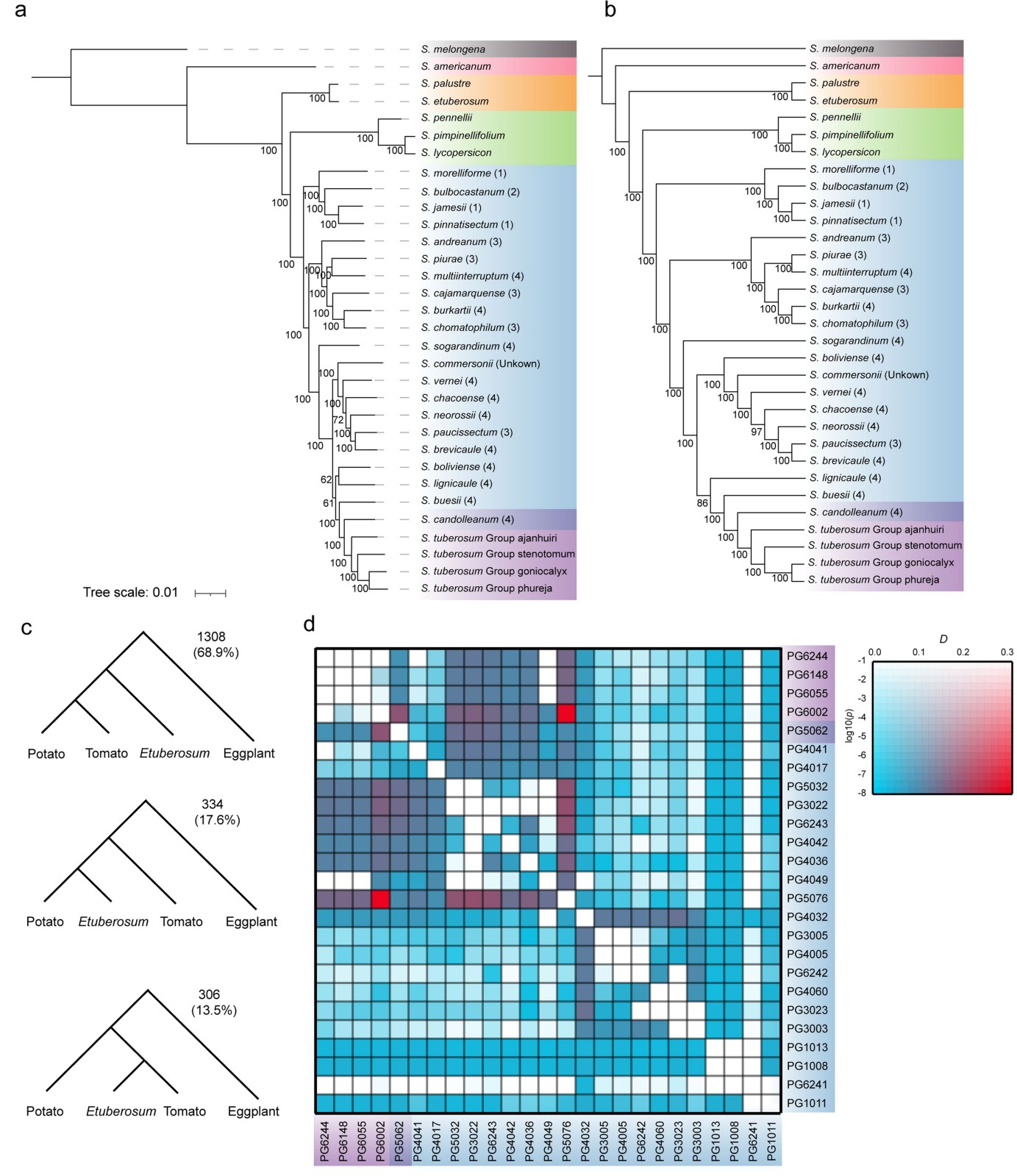

**Extended Data Fig. 3 | Phylogenetic analysis of the 32 representative accessions. a**, Maximum likelihood super-matrix tree based on 3,971 single-copy ortholog genes. The scale bar represents branch lengths, which corresponds to the mean number of substitutions per site in the alignments. **b**, Coalescent tree based on 3,971 single-copy ortholog genes, accounting for ILS. **c**, The proportion of different tree topologies among 1,899 non-overlapping window trees. **d**, Heat map of the most significant *D* scores observed between two given potato accessions (P2 and P3) across all possible individuals in P1 species. *D* scores and log$_{10}$(*p*) values are shown in different colour schemes. *Lycopersicon*, *Etuberosum*, *S. americanum* and *S. melongena* are used as an outgroup. The *P* values are calculated using a standard block-jackknife procedure as in ref. [28].

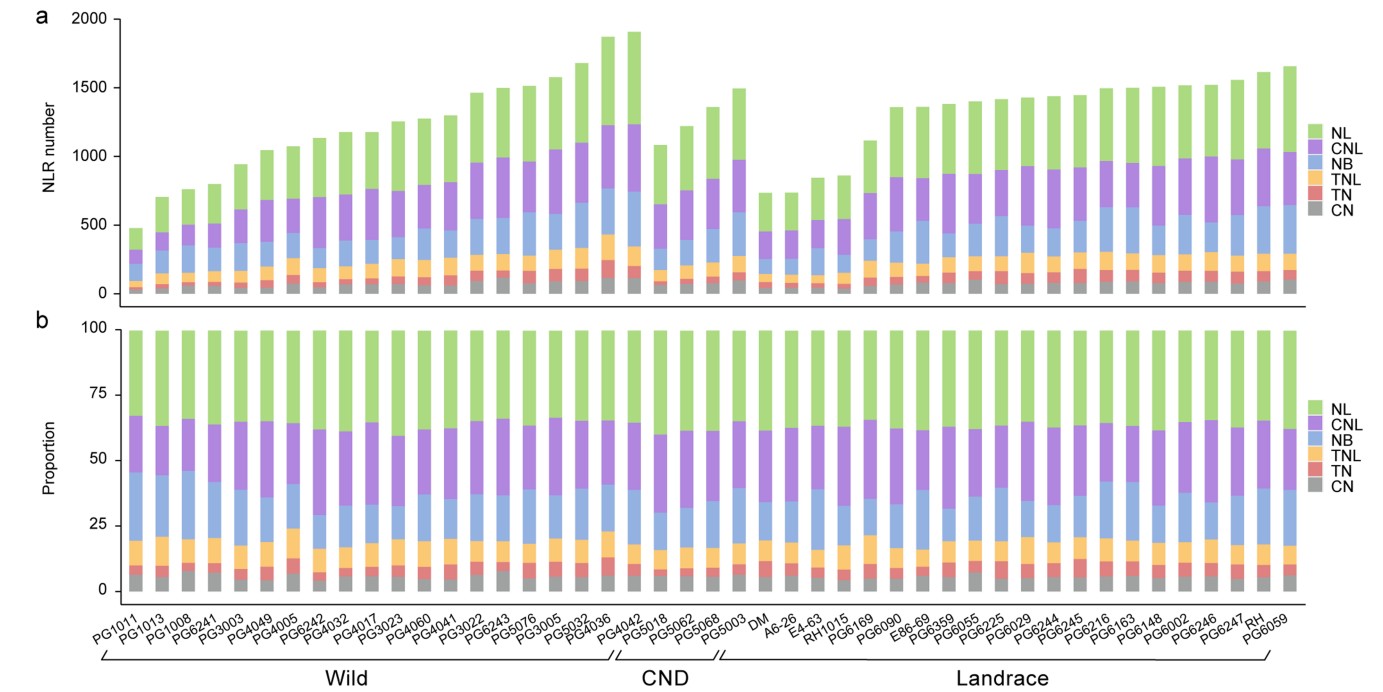

**Extended Data Fig. 4 | Landscape of NLRs in the potato genome. a**, NLR copy number in six canonical classes. NL: NB-LRR, CNL: CC-NB-LRR, NB: NB domain only, TNL: TIR-NB-LRR, TN: TIR-NB, CN: CC-NB. **b**, Proportion of each NLR class in 45 potato genomes.

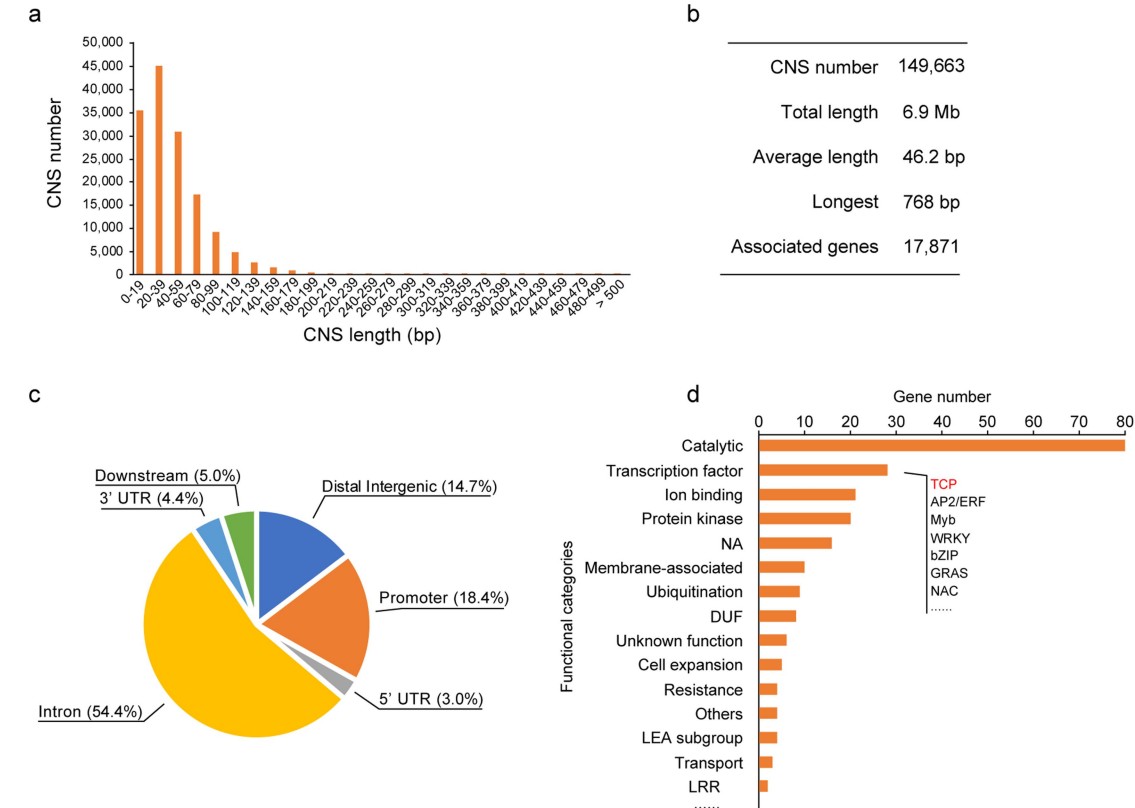

**Extended Data Fig. 5 | Features of potato-specific CNSs and categories of 229 candidate genes. a**, CNS length distribution. **b**, Summary of CNSs in potato. **c**, Pie chart shows the distribution of CNSs in potato genome. **d**, Functional categories of 229 CNS-associated potato core genes displaying stolon or tuber predominant expression.

a

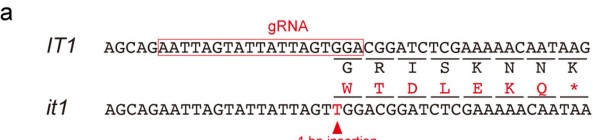

```
                     gRNA
IT1    AGCAG AATTAGTATTATTAGTGGA CGGATCTCGAAAAACAATAAG
                                 G   R   I   S   K   N   N   K
                                 W   T   D   L   E   K   Q   *
it1    AGCAGAATTAGTATTATTAGTTGGACGGATCTCGAAAAACAATAA
                                 ▲
                            1 bp insertion
```

b

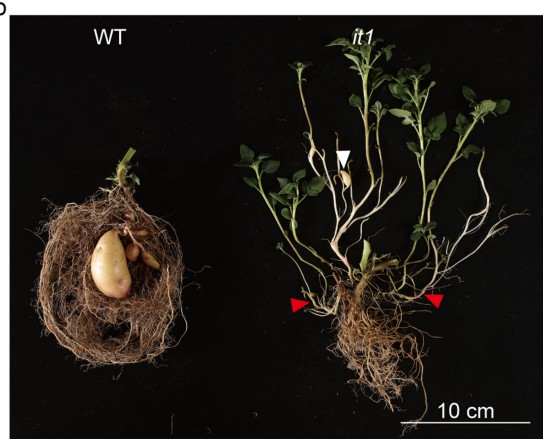

**Extended Data Fig. 6 | Phenotypes of the *it1* knockout mutant. a**, The *IT1* CRISPR/Cas9 knockout mutant. **b**, The *it1* mutant shows an impaired tuberization phenotype. The main stems were removed. Red arrows indicate *it1* stolons that convert to branches. The white arrow shows a small tuber formed on *it1*.

a

b

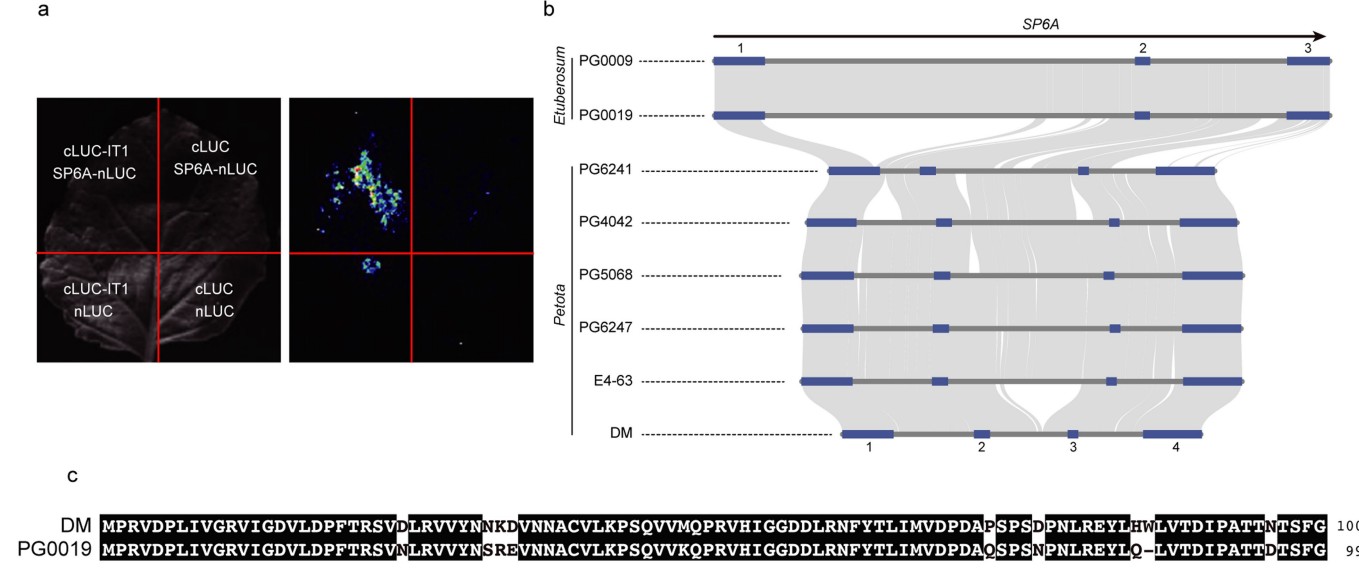

c

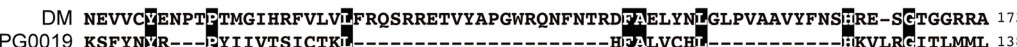

```
DM      MPRVDPLIVGRVIGDVLDPFTRSVDLRVVYNNKDVNNACVLKPSQVVMQPRVHIGGDDLRNFYTLIMVDPDAPSPSDPNLREYLHWLVTDIPATTNTSFG 100
PG0019  MPRVDPLIVGRVIGDVLDPFTRSVNLRVVYNSREVNNACVLKPSQVVKQPRVHIGGDDLRNFYTLIMVDPDAQSPSNPNLREYLQ-LVTDIPATTDTSFG  99

DM      NEVVCYENPTPTMGIHRFVLVLFRQSRRETVYAPGWRQNFNTRDFAELYNIGLPVAAVYFNSHRE-SGTGGRRA 173
PG0019  KSFYNYR---PYIIVTSICTKL--------------------HFALVCHL-----------HKVLRGITLMML 138
```

d

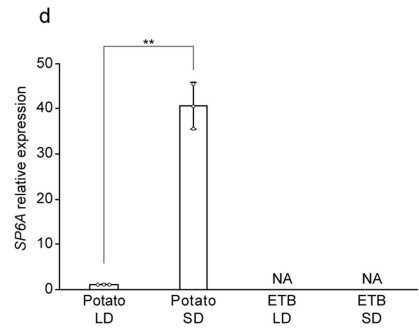

**Extended Data Fig. 7 | Interaction, sequence synteny and expression of *SP6A*. a**, Interaction between IT1 and SP6A revealed by the firefly luciferase complementation imaging assay. Three independent experiments are performed. **b**, Synteny plot of *SP6A* genomic sequences from representative *Etuberosum* and potato species. Blue boxes indicate the exons of *SP6A*, and grey blocks show collinear regions among these genomes. **c**, The protein sequence alignment of SP6A between DM and PG0019. **d**, The *SP6A* expression in potato (E4-63) and *Etuberosum* (PG0019) leaves at ZT4. ** *P*-value = 1.59e-04 in two-sided Student's *t*-test. ETB: *Etuberosum*. LD: long-day. SD: short-day. Data presented in mean ± SD, *n* = 3. Three independent experiments are carried out.

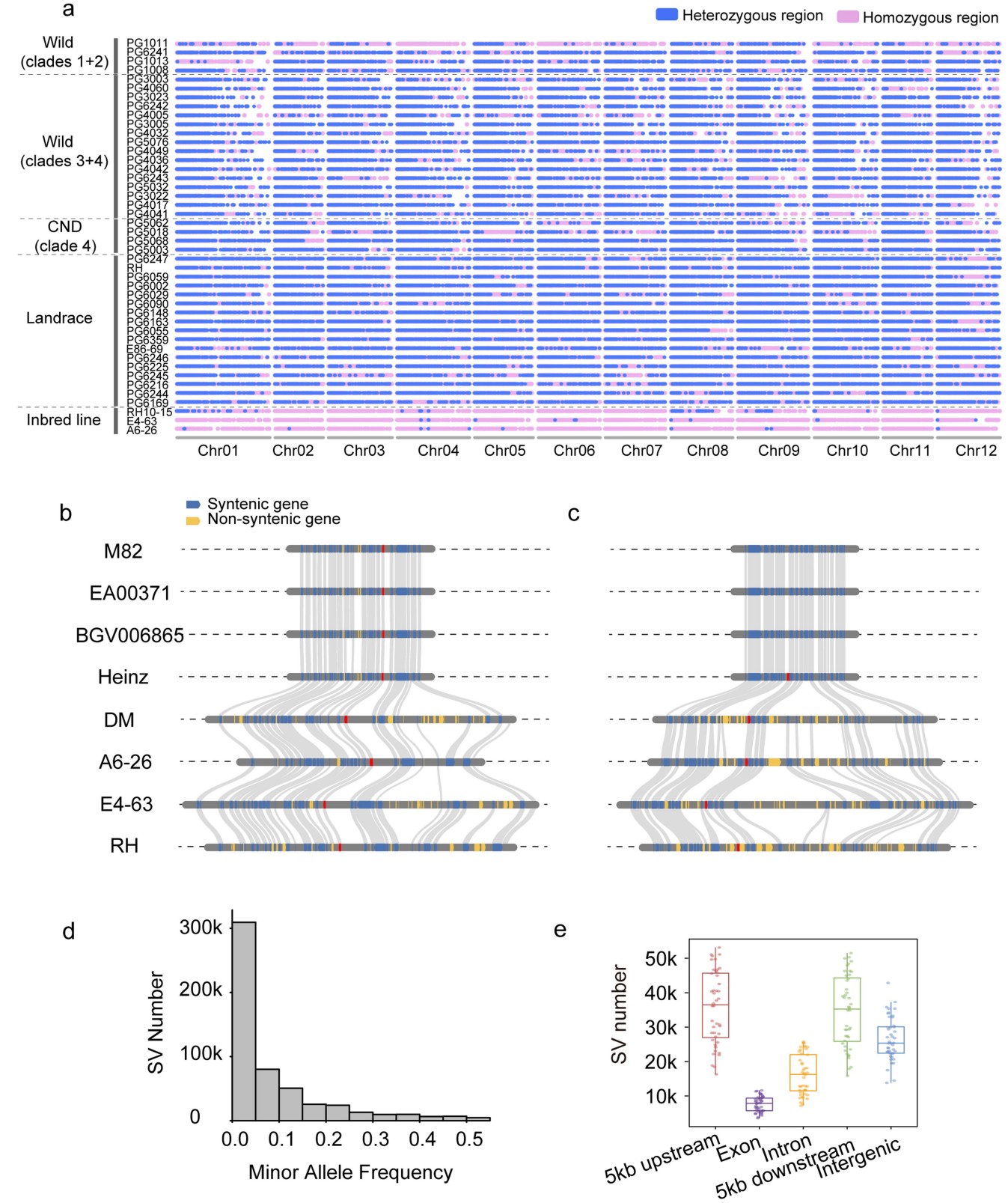

**Extended Data Fig. 8 | Genome-wide sequence variation of the 44 potato genomes. a**, Genomic architecture of heterozygosity distribution in 44 diploid potato genomes revealed by alignment to the DM reference genome; heterozygous (blue) and homozygous (pink) regions, respectively. **b**, Local synteny (DM chr12: 53.57–54.31 Mb) illustration surrounding the *GLYCOALKALOID METABOLISM 4* (*GAME4*) locus. **c**, Local synteny (DM chr01: 0.65–1.13 Mb) illustration surrounding *FLAVIN-BINDING KELCH REPEAT F-BOX*

*PROTEIN* (*FKF1*). Genes from four potato landraces (DM, A6-26, E4-63, and RH) and four cultivated tomatoes (Heinz 1706, BGV006865, EA00371 and M82) are shown. **d**, SV allele frequency among the 44 potatoes. **e**, Number of SVs localized at regulatory, genic and intergenic regions. The upper and lower edges of the boxes represent the 75% and 25% quartiles, the central line denotes the median and the whiskers extend to 1.5 × IQR. The number of genomes investigated in each category is 44.

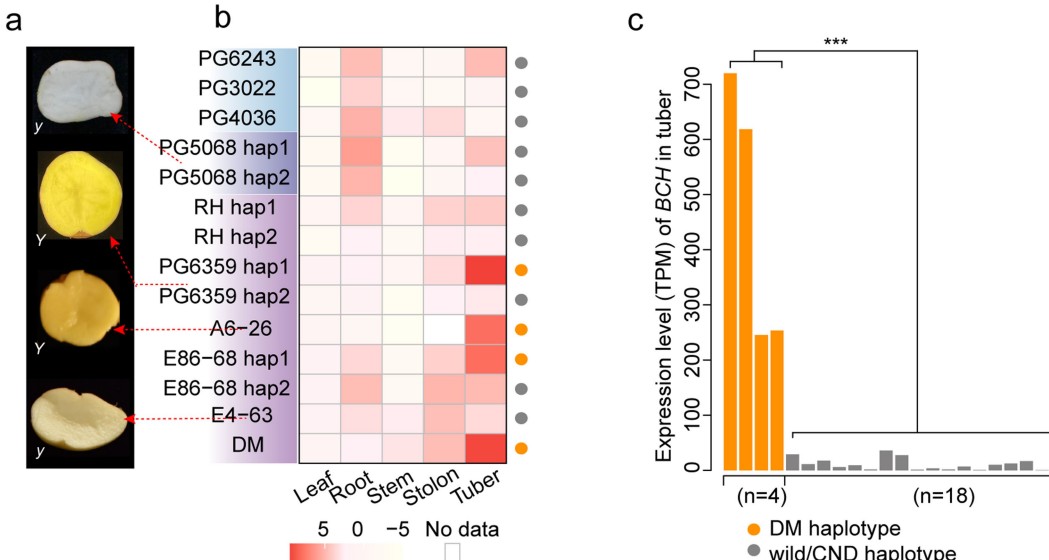

**Extended Data Fig. 9 | Association between tuber flesh colour, *BCH* expression level and the presence of the 5.8-Mb inversion. a**, Phenotypes of tuber colour for accessions E4-63, A6-26, PG6359 and PG5068. **b**, Expression level (log₂TPM) of *BCH* in five tissues of wild, CND and landrace accessions/haplotypes. Orange dot denotes DM haplotype, and grey dot denotes wild/CND haplotype. DM haplotype: accessions without the inversion; Wild/CND haplotype: accessions carrying the inversion. **c**, Expression level (TPM) of *BCH* in tubers of 22 accessions/haplotypes, including 4 DM haplotypes, and 18 wild/CND haplotypes. *** *P*-value = 1.462e-07 in two-sided Student's *t*-test.

# Reporting Summary

## Statistics

For all statistical analyses, confirm that the following items are present in the figure legend, table legend, main text, or Methods section.

| n/a | Confirmed | |
|---|---|---|
| ☐ | ☒ | The exact sample size (*n*) for each experimental group/condition, given as a discrete number and unit of measurement |
| ☐ | ☒ | A statement on whether measurements were taken from distinct samples or whether the same sample was measured repeatedly |
| ☐ | ☒ | The statistical test(s) used AND whether they are one- or two-sided *Only common tests should be described solely by name; describe more complex techniques in the Methods section.* |
| ☒ | ☐ | A description of all covariates tested |
| ☒ | ☐ | A description of any assumptions or corrections, such as tests of normality and adjustment for multiple comparisons |
| ☐ | ☒ | A full description of the statistical parameters including central tendency (e.g. means) or other basic estimates (e.g. regression coefficient) AND variation (e.g. standard deviation) or associated estimates of uncertainty (e.g. confidence intervals) |
| ☐ | ☒ | For null hypothesis testing, the test statistic (e.g. *F*, *t*, *r*) with confidence intervals, effect sizes, degrees of freedom and *P* value noted *Give P values as exact values whenever suitable.* |
| ☒ | ☐ | For Bayesian analysis, information on the choice of priors and Markov chain Monte Carlo settings |
| ☒ | ☐ | For hierarchical and complex designs, identification of the appropriate level for tests and full reporting of outcomes |
| ☒ | ☐ | Estimates of effect sizes (e.g. Cohen's *d*, Pearson's *r*), indicating how they were calculated |

*Our web collection on statistics for biologists contains articles on many of the points above.*

## Software and code

Policy information about availability of computer code

| Data collection | No software was used to collect data. Sequencing platforms used to generate the raw data are listed as followed: PacBio Sequel II, PacBio RS II,  Illumina HiSeq X Ten, DNBSEQ-T7. |
|---|---|
| Data analysis | We used publicly available and appropriately cited software in the Methods. No commercial software and code were used in this study. Software are listed as follows: BWA 0.7.5a-r405, SAMtools v1.9, bcftools v1.9, IQ-TREE v2.0.6, GenomeScope2.0, hifiasm v0.13, purge_dups (v1.01), CANU v1.8, Pilon v1.23, juicer v1.5, 3d-dna v180922, EDTA v1.9.4, RepeatMasker v1.332, HISAT2 (v2.0.1-beta), StringTie (v1.3.3b), BRAKER2 v2.1.6, AUGUSTUS (v3.4.0), GeneMark-ET (v3.67_lic), StringTie (v1.3.3b), MAKER2 (v2.31.11), SNAP (v2013-02-16), InterProScan 5.34-73.0, BUSCO v4.1.4, BLASTP (v2.2.30+), OrthoFinder (v 2.5.2), PanGP (v1.0.1), ParaAT (v2.0), ProgressCactus (v1.2.3), R v4.0.3, PHAST v1.5, DIAMOND (v2.0.6.144), ASTRAL (v5.7.8), MUMmer v4.00rc1, MAFFT (v7.471), BedTools (v2.29.2), PAML v4.9, Dsuite (v0.4 r28), SVIM (v1.4.2), Assemblytics (v1.2.1), SURVIVOR (v 1.0.7), DendroPy (no version), ragtag (v2.1.0), SyRI (v1.4), minimap2 v2.21-r1071, NLR-annotator (v0.7), PRGdb 3.0, RGAugury (v2.2), MView (v1.67), ChIPseeker (v1.24.0) and pyGenomeTracks (v3.6). |

For manuscripts utilizing custom algorithms or software that are central to the research but not yet described in published literature, software must be made available to editors and reviewers. We strongly encourage code deposition in a community repository (e.g. GitHub). See the Nature Portfolio guidelines for submitting code & software for further information.

## Data

Policy information about availability of data

All manuscripts must include a data availability statement. This statement should provide the following information, where applicable:
- Accession codes, unique identifiers, or web links for publicly available datasets
- A description of any restrictions on data availability
- For clinical datasets or third party data, please ensure that the statement adheres to our policy

All PacBio sequence data, transcriptome data, Hi-C data in this study have been deposited at the National Center for Biotechnology Information (NCBI) Sequence Read Archive (SRA) with BioProject accession number PRJNA754534 and the National Genomics Data Center (NGDC) Genome Sequence Archive (GSA) with BioProject number of PRJCA006011. Genome assemblies, annotation, sequence variation, gene expression for the 46 accessions and the genotype information of 432 lines used for sample selection have been hosted in database the Pan-potato Database (http://solomics.agis.org.cn/potato/, http://218.17.88.60/potato/). Publicly available sequencing data were downloaded from the NCBI with BioProject number of PRJNA641265, PRJNA573826, PRJNA378971, PRJNA394943 and PRJNA766763.

# Field-specific reporting

Please select the one below that is the best fit for your research. If you are not sure, read the appropriate sections before making your selection.

☒ Life sciences ☐ Behavioural & social sciences ☐ Ecological, evolutionary & environmental sciences

For a reference copy of the document with all sections, see nature.com/documents/nr-reporting-summary-flat.pdf

# Life sciences study design

All studies must disclose on these points even when the disclosure is negative.

| | |
|---|---|
| Sample size | For pan-genome construction, 44 representative potato accessions were used. These accessions were selected based on their phylogenetic relationship and represented genetic diversity in the potato germplasm. |
| Data exclusions | No data was excluded. |
| Replication | Three biological replicates with three technical replicates were used in the qRT-PCR experiment. Three biological replicates were conducted in the yeast-two-hybrid assay. Three independent transgenic knock-out lines were generated for IT1. All replications were successful and were used. |
| Randomization | Randomization does not directly apply to the genome sequencing and assembly. |
| Blinding | Blinding does not apply to this study, as the study focuses on comparative genomics and blinding is not necessary. |

# Reporting for specific materials, systems and methods

We require information from authors about some types of materials, experimental systems and methods used in many studies. Here, indicate whether each material, system or method listed is relevant to your study. If you are not sure if a list item applies to your research, read the appropriate section before selecting a response.

## Materials & experimental systems

| n/a | Involved in the study |
|---|---|
| ☒ ☐ | Antibodies |
| ☒ ☐ | Eukaryotic cell lines |
| ☒ ☐ | Palaeontology and archaeology |
| ☒ ☐ | Animals and other organisms |
| ☒ ☐ | Human research participants |
| ☒ ☐ | Clinical data |
| ☒ ☐ | Dual use research of concern |

## Methods

| n/a | Involved in the study |
|---|---|
| ☒ ☐ | ChIP-seq |
| ☒ ☐ | Flow cytometry |
| ☒ ☐ | MRI-based neuroimaging |

