## [Peer Review File · Nature]

Manuscript Title: Genome evolution and diversity of wild and cultivated potatoes

Reviewer Comments & Author Rebuttals

Reviewer Reports on the Initial Version:

Referees' comments:

Referee #1 (Remarks to the Author):

Tang et al. report on the construction and initial analysis of a genus-wide pan-genome of potato. Their results are impressive. They have assembled chromosome-scale genome sequences for tens of species using a state-of-the-art method (PacBio HiFi + Hi-C). The genomes helped them unravel the relationships between species and glean insights into the evolution of tuberization. Their most exciting accomplishment is the discovery of structural variants affecting agronomic traits: one inversion tightly linked with a flesh color gene and presence-absence variation for a regulatory region of a gene involved in tuberization. Overall, the paper is well-written and easy to follow. The display items are of high-quality and support the results. I think this study merits publication in Nature. It will become a widely cited community resource and illustrates the power of genus-wide pan-genomics, an approach that is currently being followed in many other crops, but which has not yet resulted in a paper of the depth and breadth as Tang et al.'s effort. I have some requests for more explanation / technical validation of the assembly methods and suggestion how to improve the clarity of the writing.

Major points:

1. The author do not clearly describe how they deal with heterozygosity in the assembly process. The relevant information is hidden in technical jargon (PTG, ATG, etc.). My understanding is that the author did not attempt chromosome-scale diploid genome assembly, but constructed pseudo-haploid chromosomes. This is a defensible approach given that (i) there is no standard method for diploid genome assembly, (ii) the authors' analyses do not rely on phased diploid genomes and (iii) the authors provide also alternate contigs omitted from the pseudo-haploid assembly. The authors should describe the rationale for using pseudo-haploid assemblies in terms intelligible to a geneticist.
2. It is important to assess how well the haploidization steps works. How many homologous regions are not found by the purge_dup algorithm and thus retained as artificial duplications in the pseudo-haploid assembly?
3. I would like to see more technical validation of the chromosome-scale scaffolding. To which extent was manual curation performed? Ideally, high-resolution (!) Hi-C contact matrices should be shown for all assembled genomes as supplementary files or deposited under a DOI. Which measures did the authors take to distinguish true structural variants from possible assembly artifacts. This is of particular relevance for the inversion linked to the flesh color gene.

Minor points:

- L. 29: is a paradigm shift. That's too optimistically phrased. The recent paper by the Huang group on hybrid potato is indeed impressive. Their approach has the potential to usher in a paradigm shift in potato breeding, but this potential has yet to be realized.
- L. 35: wide diversity: without quantification this is not very meaningful, it may be better to simply say ... accessions from the Solanum section Petota.
- L. 44 controlling a nutritional trait: I'd prefer a more specific phrasing spelling out the trait
- L. 56 facilitate more rapid -> speed up
- L. 60 empower: that's a complicated way for saying potato reproduce vegetatively by tubers
- L. 60 its: it's not quite clear to what 'it' refers to. Given that reproduction is the *raison d'être* for tuberization, the logic of this sentence looks a bit muddled
- L. 77 it will be more informative to report coverage instead of leaving the reader to figure out genome size and do the calculations themselves
- L. 79 introduce the terminology
- L. 82 Hi-C: at least spell out the abbreviation, possibly refer to relevant papers on Hi-C scaffolding (Burton et al./Kaplan et al.)
- L. 97/98: I'm OK with the conclusion, but the logic is not clear, and I don't agree with the premise. Seeing a plateau curve ED Fig. 1d is in the eye of the beholder. I don't see one. It's probably not necessary to do a full-fledged saturation analysis (whose conceptual basis is also vague without knowing the universe). Some more cautious phrasing would do.
- L. 100 soft/shell: terminology in need of introduction
- L. 103: Interpro protein: technical jargon, better: protein with domains annotated in the Interpro database
- L. 108: Ka/Ks is not well-defined in intraspecific comparisons (see PMID: 19081788). Your panels is a mixture of samples for the same and from different species. Ka/Ks makes only in between species
- L. 122: Given the importance later in the text, a definition of stolon is appreciated
- L. 126: long reads: be more specific, CLR and HiFi are very different.
- L. 140: This sentence doesn't fit into the context. What are the traits? Do you mean to compare single-locus vs. genome-wide trees?
- L. 156: extensive interspecific hybridization: I'm not well-read in the potato literature. But I expect this phenomenon may be observed before with different flavors of molecular markers. If so, the relevant papers should be cited
- L. 162: rapid decay: I think this misses the point. Selfing crops were often domesticated from selfing wild progenitors. So heterozygosity is low in either taxon.
- L. 165: increased heterozygosity: relative to what? To selfers, yes, but that's almost self-evident. Wild vs domesticated? It's not clear to me why domestication per se should increase heterozygosity.
- L. 168 ATG/PTG: define terminology
- L. 174 Can you rule higher homozygosity due to a very recent bottleneck on a population genetic time scale, rather a change of mating system in phylogenetic time?
- L. 181: I think you refer to *deleterious* variants. The accumulation of genetic variants ultimately traces back to incoming cosmic radiation, on which clonal propagation has no bearing.
- L. 189: SL4.0 technical jargon
- L. 191: This conclusion is odd. In ED Fig. 4, everything looks quite collinear. Your phrasing is vague. A concrete scenarios would be the accumulation of structural variants that dramatically reshuffle

genes, but ED Fig. 4 shows nothing of the kind. It needs to be spelled out was you mean by synteny (at which scale?) and by substantial loss.

L. 193: Founder is a term from pedigree breeding. Is it well-defined for landraces?

L. 195: The sentence stands without the relative clause, which I think makes the implicit assumption that you are talking about *deleterious* variants. One does not purge something beneficial.

L. 204-207: I'd like to see some more technical validation of the inversions, e.g. by showing Hi-C contact matrices or genetic maps. If the inversions are contained in single HiFi contigs, that's also strong support their validity.

L. 211: left The genome knows no left or right. Distal and proximal are better terms.

L. 217-219: *is the causative variant* This conclusion is too strong. This can only be proven by re-creating the inversion by gene editing. Rephrase in a more cautious way.

L. 222 hampering: It's conceivable that inversions create beneficial super-genes that actually boost crop improvement.

L. 228: I don't think "purposefully" is needed here. The purpose (hybrid breeding) is understood. Maybe rephrase: select parental lines with optimal nutritional profiles.

L. 232: The logic of this sentence is a bit muddy. How about: Clonal propagation gave rise to tuber-borne diseases against which potato has evolved a repertoire of novel resistance genes.

L. 240: The switch to tomato needs a better motivation. At first, I had the impression that you were almost literally comparing apples and oranges.

L. 250: monoploid: see my comment above about QC on how well "haploidization" in hifiasm works

L. 262-267: This is a short paragraph on a complex analysis comparing two species unrelated to potato. An obvious technical concern if the Ipomoeae assemblies are good enough to support this analysis

L. 284-285: "whole genome alignment considering evolutionary distance" This is vague. It leaves the reader in the dark about what was actually done. The Methods are more informative, but a better summary is needed here.

L. 349: To my understanding, "it has not eluded us" is a tongue-in-check way of introducing some obvious fact. It does not fit here.

L. 361: hybrid designation: I'm not sure what this means.

L. 495: Instead of saying "standard" refer to the relevant protocol.

Referee #2 (Remarks to the Author):

In this manuscript entitled "Genome evolution and diversity of wild and cultivated potatoes", Tang and colleagues assemble and analyze high-quality genomes of 44 diploid potatoes, including both wild and cultivated accessions, with the aim of improving our understanding of the potato genome and tuber evolution. Assembling and analyzing this number of diploid, often highly heterozygous genomes is a technical feat. Many of the genomic analyses presented are largely descriptive and essentially provide a catalog of genomic variation among potato accessions. For instance, the study describes a potato pan-genome, identifies structural variants and potato-specific conserved non-coding regions, and describes the distribution of homozygous and heterozygous genome segments in the sequenced potato accessions, as well as the types and numbers of resistance genes and the distribution of syntenic segments. The manuscript also presents analyses of the phylogenetic relationships among potato and related species, as well as functional work, including the

identification of a likely tuber initiation gene using CRISPR-Cas9-based genome editing. The manuscript thus presents a substantial body of work aiming to document genomic variation and provide insights into the genetic basis of tuber formation in potatoes. The manuscript is mostly clearly written and the figures are clear.

Although genomic diversity in tuber-bearing *Solanum* has recently been described (e.g. Hardigan et al. 2017. PNAS), this manuscript improves on earlier studies by basing its analyses on highly contiguous genome assemblies. The study relies on the latest methods for long-read sequencing and assembly, and the resulting genome assemblies seem to be overall of high quality. That being said, the evolutionary genomic analyses undertaken are not very novel, and do not yield major novel biological insights. So, while the genomic resources generated will be very valuable for further work on this important crop system and also provides opportunities for evolutionary genomic analyses, the results presented here do not stand out in terms of originality and significance. While genomic analyses seem well executed for the most part, there is frequently a disconnect between the conclusions drawn and the conclusions that would be possible to support based on the conducted analyses. This is particularly the case for the analyses of genomic patterns, where the conclusions drawn in the manuscript are often not supported by the analyses or the data, but also occurs in connection with some of the functional analyses. Furthermore, some analyses are not described in sufficient detail, resulting in poor reproducibility, and statistical tests are sometimes not correctly described.

In sum, while it is certainly of interest to crop breeders to have access to these genome assemblies and the catalogue of genomic variation, I am not convinced that the main conclusions are supported, nor that the study will be of sufficient interest for a very broad and general readership. Below, I outline both major and minor concerns in more detail.

Major comments

1. Conclusions on hybridization are not supported by analyses

The phylogenetic reconstructions done to clarify the position of *Etuberosum* are nicely done, and I appreciate that the authors used a combination of super-matrix analyses and coalescent-based analyses to reconstruct a species tree. However, the authors then go on to conduct analyses in 100 kb non-overlapping windows, and use the widespread gene tree-species tree discordance to argue for a "complex history of potato evolution including extensive interspecific hybridization" (Lines 151-154). There are several problems with this interpretation. The first one is methodological – based on the methods it seems that for this analysis, all non-coding regions were kept, and there is no mention of filtering out repetitive non-coding regions. This is problematic, as erroneous SNP calls in repetitive regions are common and can result in gene tree discordance even in the absence of interspecific hybridization (or for that matter, incomplete lineage sorting). If repeats were not already filtered, the analysis should be redone after doing so. However, the second problem is possibly inherent to the system being studied. When analyzing genomic data from a set of closely related species, it is not necessarily unexpected to observe gene tree discordance, and the results that are presented do not allow the authors to draw conclusions on the causes of the discordance. To be able to draw such conclusions much more rigorous analyses would be needed, taking into account that gene trees often do not have the same topology as the species tree when internodal

branches are short relative to the effective population size (as shown by Pamilo and Nei 1988). As the authors do not present any estimates of divergence time (e.g. based on synonymous divergence), nor provide information on intraspecific polymorphism at neutrally evolving sites (which is related to the effective population size), it is difficult to develop an intuition for how rampant incomplete lineage sorting would be expected to be in this case. However, it is clear that based on the presented analyses, it is not possible to draw any firm conclusions on the frequency of interspecific hybridization.

2. Conclusions on reproductive mode are not supported by analyses

The authors investigated substantial variation in the distribution of homozygous and heterozygous segments in the sequenced genomes and drew the conclusion that "These results suggest that the reproduction strategy incorporating both tubers and seeds of potato in nature resulted in the heterogeneous genomic architecture" (lines 172-174).

It is unclear how the authors can assess the impact of the complex reproduction strategy here, as there is no control group with a different reproduction strategy included in the analyses, and it is also not clear why the observed variation in the distribution of heterozygous segments is seen as remarkable.

Furthermore, the authors state (on lines 169-172) that "We also identified eight near-homozygous chromosomes in *S. morelliforme* (PG1011, Chr1, 2, 4, 6, 7, 10, 11, and 12) (Extended Data Fig. 3a), which is a self-incompatible species suggesting a recent event of conversion from self-compatibility to self-incompatibility in this accession."

It is not uncommon that even self-incompatible species can be subject to inbreeding (e.g. biparental inbreeding), which could result in homozygous segments. Therefore, observing a self-incompatible species with high homozygosity cannot be used to conclude that it was previously self-compatible. Given the evolutionary rarity of transitions from self-compatibility to self-incompatibility, the authors' claim is extraordinary, and much stronger empirical evidence (ideally on the detailed genetic basis of potentially re-evolved self-incompatibility) would be required to draw such a far-reaching conclusion.

Given the lack of a control group and the insufficient support for the conclusions drawn, I would recommend to exclude the analyses of heterozygosity from the main text altogether, or to improve the analyses by including a control group.

3. Lack of evolutionary context for analyses of synteny

The authors use multi-way alignments to compare synteny among sequenced genome of cultivated tomato vs cultivated potatoes. They document "a substantial loss of synteny in cultivated potatoes, probably due to the elevated level of genetic divergence by tuber propagation and relatively high founder diversity¹⁵." First, this analysis demonstrates the utility of a control group – it is very nice that the authors compared their results to those for cultivated tomatoes. However, assembly errors particularly in heterozygous genomes could affect inference of synteny, so it would be important to document that the assemblies of the contrasted groups are similar in quality. Assuming that the results are robust to technical artefacts, in order to fully interpret these results, more information on the level of divergence among the compared tomato accessions relative to the divergence among the potato accessions is still required. The tomato genomes analyzed are not included in any phylogeny shown, so one cannot rule out higher synteny just due to more recent divergence among

the tomato accessions analysed. The authors should provide more evolutionary context to facilitate interpretation of these results.

4. Causal nature of inversion not supported

The authors identify a 5.8 Mb inversion close to the Y locus that determines tuber flesh colour. After comparing haplotypes and expression data for accessions with white and yellow tuber colour, the authors conclude that "the 5.8-Mb inversion is the causative variant that provides a new promoter to activate the downstream of BCH gene". This statement is not strictly supported, as genetic modifications would be needed to test this hypothesis. For now, it is not possible to rule out sequence differences in the 1.5-2 kb next to the BCH gene but outside of the inversion. For that reason, this conclusion should be rephrased.

Comments on statistical analysis/description of methodology

5. Data and analyses underlying selection of accessions is not sufficiently described. The authors state that their choice of accessions was based on phylogenetic relationships of 432 accessions (lines 71-73) and refer to Supplementary Table 1 and Supplementary Figure 1. The methods states that all these accessions were sequenced using Illumina technology, but Supplementary Table 1 does not list Illumina data for all of those accessions. There is no description of how the data were processed to identify SNPs and how those SNPs were filtered, how many SNPs were analysed, how the phylogeny was generated (no software or sequence evolution model is mentioned), and bootstrap values are missing so there is no information on the support of different branches. Furthermore, it is not possible to connect Supplementary figure 1 to Supplementary table 1 as there are no labels on the taxa in the phylogeny. It is also not clear from the Reporting Summary whether the Illumina data is made available through NCBI. In short, this part of the analysis is not reproducible based on the information provided.

6. In the pan-genome analysis, the authors show that Ka/Ks values are lower for core genes than for accessory genes (called shell and accession-specific) and that expression was higher for core genes. This is a result that is in line with expectations if core genes are under stronger functional constraint, perhaps in part due to their higher expression levels and due to the inclusion of "housekeeping" genes in this category. To fully report how they arrived at this result, however, the authors would need to provide more information on how Ka/Ks estimation was done. Specifically, the KaKs_Calculator run by ParaAT can estimate Ka/Ks using a number of different methods, and here one would need some more information on which one was used. It is also not clear what is meant by "50 different combinations of Ka/Ks within each cluster were computed".

7. Statistical treatment or notation unclear

With respect to the pan-genome analysis, it is unclear how the statistical comparison of gene clusters were done, specifically regarding the data presented in Extended data figure 1 f and g. The letter labels above each category in these graphs implies that a post-hoc test has been done after a significant Kruskal-Wallis test but the figure legend states "Multiple comparison was performed using Kruskal-Wallis test with $\alpha = 0.001$ ". As the Kruskal-Wallis test does not compare groups in a pairwise manner, more information is needed to support the figure. What specific post-hoc multiple comparison test was done, with what settings? The text in methods (lines 565-567) is not sufficient.

Inconsistent regarding statistical notation is also used regarding reported correlations (line 82-85, Extended Data Fig. 1c), please use r or R consistently for Pearson's correlation coefficient.

7. How the phylogenetic analyses were done is not described in sufficient detail in the methods. For instance, it is stated that "To build a super-matrix species tree, amino-acid sequences of representative annotated gene models were first extracted from genomes of 44 PTGs, DM v6.112, 24 tomato accessions (see Supplementary Table 1 for more details)^{26,42,74,75}, two *Etuberosum* species and eggplant⁸¹."

Supplementary Table 1 contains more information on the accessions but not on the "representative annotated gene models" – how were they selected, and how many single copy clusters were included in the IQ-TREE analysis?

Minor comments on data availability and presentation

8. Data availability unclear

It is not clear whether all data, including Illumina whole-genome resequencing data used for accession choice, has been uploaded to an appropriate repository such as NCBI. The Data part of the Reporting Summary statement only lists PacBio data, transcriptome data, and Hi-C data as having been deposited there, and a reference to a custom database with unknown access terms (the database is not currently accessible) is provided for "genome assemblies, annotation, sequence variation, gene expression". Note however that the Methods states that all sequence data has been uploaded to NCBI – this is currently unclear.

9. Lines 95-97 "Pan-genome size increased when incorporating more genomes and reached a plateau when n was close to 40 (Extended Data Fig. 1d), suggesting that our collection represents the broad biodiversity of potato."

Rephrase to "our collection captures the shared gene content of potato" as not all aspects of genetic diversity or biodiversity are captured by variation in gene content.

10. In the abstract on (line 31-32) the following sentence should be rephrased: "However, it remains under-investigated on the genome evolution and diversity of wild and cultivated potato, limiting the utilization of rich variation in the potato gene pool."

Referee #3 (Remarks to the Author):

A. Summary of the key results

Using new sequencing technologies and de novo genome assembly software, authors delivered pan-genome of heterozygous potatoes covering diploid and wild species. The study presents new insight into the genetics and evolution of tuber formation in plant species by providing comprehensive genome and transcriptomes of *Petota* as well as a *Lycopersicon* and *Etuberosum* accessions. It also sheds some light on structural variation affecting the expression of genes encoding for yellow flesh, which is an important nutritional trait and important target for genetic improvement. These genomic resources improve our understanding of interspecific variation and genome evolution in potatoes. They also provide the foundation for genomics-informed potato breeding and genetic resources

management.

B. Originality and significance: if not novel, please include reference

The study extends our knowledge of potato evolution by analysing genome of accessions from *Solanum* species belonging to clades 1-4. The results are of immense interests to potato breeders and gene bank curators to guide efficient breeding and genetic resources management strategies.

C. Data & methodology: validity of approach, quality of data, quality of presentation

All data were thoroughly reviewed and no flaw was detected. However, since I am not an expert in bioinformatics and taxonomy, some details may have escaped me.

D. Appropriate use of statistics and treatment of uncertainties

All data were thoroughly reviewed and no flaw was detected. However, since I am not an expert in bioinformatics and taxonomy, some details may have escaped me.

E. Conclusions: robustness, validity, reliability

The panel of germplasm used included tuberizing and non-tuberizing species from the Solanaceae family, as well as outbreeders and inbreeders from a wide range of geographical distribution. Most of the discussion is centered around the impact of tuberization on potato genomes and evolutionary mechanisms, whereas a possible role of ecological distribution and interspecific hybridizations or introgressions was not as much covered. Overall, conclusions are valid and solidly founded.

F. Suggested improvements: experiments, data for possible revision

A minor revision is suggested to include discussion on possible role of species geographical distribution and breeding behaviour on potato genome evolution in the context of the present study.

G. References: appropriate credit to previous work?

Most relevant references were included.

H. Clarity and context: lucidity of abstract/summary, appropriateness of abstract, introduction and conclusions

The quality of the manuscript is excellent.

Author Rebuttals to Initial Comments:

We greatly appreciate the Reviewers' comments to improve our manuscript. To address these comments, we have adjusted the logic of the results section, performed additional analyses and carefully revised the manuscript. Below we provide a point-by-point response to the reviewers' comments and indicate how we have modified the manuscript. In addition, all revisions regarding the reviewers' concerns have been highlighted in **yellow** background in the revised manuscript.

Referee #1 (Remarks to the Author):

Tang et al. report on the construction and initial analysis of a genus-wide pan-genome of potato. Their results are impressive. They have assembled chromosome-scale genome sequences for tens of species using a state-of-the-art method (PacBio HiFi + Hi-C). The genomes helped them unravel the relationships between species and glean insights into the evolution of tuberization. Their most exciting accomplishment is the discovery of structural variants affecting agronomic traits: one inversion tightly linked with a flesh color gene and presence-absence variation for a regulatory region of a gene involved in tuberization. Overall, the paper is well-written and easy to follow. The display items are of high-quality and support the results. I think this study merits publication in Nature. It will become a widely cited community resource and illustrates the power of genus-wide pan-genomics, an approach that is currently being followed in many other crops, but which has not yet resulted in a paper of the depth and breadth as Tang et al.'s effort. I have some requests for more explanation / technical validation of the assembly methods and suggestion how to improve the clarity of the writing.

Reply: We appreciate the reviewer's comments and advice. In the revised manuscript, we have provided additional details on the assembly methods, haplotig purging, evaluation metrics and interpretation, Hi-C analyses, and evolutionary analyses. These details can be found in the **Methods** and **Supplementary Figures**. We have also substantially re-written and integrated the two original sections "**Genomic architecture of heterozygous diploid potato**" and "**Extensive divergence in *Petota***" into "**Pan-genome-guided hybrid potato breeding**", with the aim to facilitate hybrid potato breeding using our pan-genome analyses. Therefore, some content regarding the reviewer's concerns have been revised or removed. We believe that this addresses the reviewer's concerns.

Major points:

1. The author do not clearly describe how they deal with heterozygosity in the assembly process. The relevant information is hidden in technical jargon (PTG, ATG, etc.). My understanding is that the author did not attempt chromosome-scale diploid genome assembly, but constructed pseudo-haploid chromosomes. This is a defensible approach given that (i) there is no standard method for

diploid genome assembly, (ii) the authors' analyses do not rely on phased diploid genomes and (iii) the authors provide also alternate contigs omitted from the pseudo-haploid assembly. The authors should describe the rationale for using pseudo-haploid assemblies in terms intelligible to a geneticist.

Reply: We are sorry about the confusion. We have improved the methods by providing more details in the revised manuscript (Lines 524-538). We have also provided a flow chart to explain the assembly process and haploidization step (**Response Fig. 1**).

The initial output of hifiasm (v0.13) yields a pair of assemblies: (1) the primary assembly (in hifiasm named p_ctg) representing a mosaic haplotype without purging; and (2) the alternate assembly (in hifiasm named a_ctg) which represents the alternate haplotype absent from the primary one¹. For diploid potato, the monoploid genome size was estimated to be 700 – 800 Mb²⁻⁴; thus, the initial p_ctg generated by hifiasm in this study (689 – 1,562 Mb) cannot be directly used for comparative analyses. We used Purge_dups to identify duplicates and evaluate the purged assemblies using the k-mer spectra method (please see the next response for more details).

Generating haplotype-resolved diploid assemblies is not the major focus of this study and, moreover, the major conclusions of our research do not rely on phased genome assemblies. We did not focus on intra-haplotype diversity either, which has been described in Zhou *et al.*, 2019 (ref⁵), in which we generated a haplotype-resolved genome assembly of a diploid potato, using HiFi, Hi-C and selfing population data.

Response Fig. 1: The workflow of genome assembly and haploidization. MTG, monoploid assembled contigs. ATG, alternate assembled contigs.

2. It is important to assess how well the haploidization steps works. How many homologous regions are not found by the purge_dup algorithm and thus retained as artificial duplications in the pseudo-haploid assembly?

Reply: We appreciate your concern regarding the haploidization steps. We have improved the description of our haploidization approach in the **Methods** in the revised manuscript (Lines 524-538). We used the k-mer spectra method⁶ to assess the raw assemblies and the purged assemblies, and we added k-mer spectra plots for all assembled genomes to **Supplementary Fig. 3**. Below, we highlight one case to explain how we assessed the haploidization steps (**Response Fig. 2**). The left plot shows the k-mer pattern of the initial assembly of a heterozygous diploid, while the right panel displays the pattern after purging $2 \times$ genomic regions. A similar pattern was observed in the vast majority of our assemblies.

Response Fig. 2: K-mer spectra assessment for raw and monoploid assemblies of an example accession. For each sample, the left plot shows the 31-mer spectra of the initial assembly, and the right plot illustrates the 31-mer spectra of the purged assembly. The x-axis indicates the multiplicity of distinct k-mers of raw HiFi reads. The colors represent the times the k-mers were present in the genome assembly. After purging, the heterozygous regions (purple) are mostly collapsed into single-copy homozygous content (red).

3. I would like to see more technical validation of the chromosome-scale scaffolding. To which extent was manual curation performed? Ideally, high-resolution (!) Hi-C contact matrices should be shown for all assembled genomes as supplementary files or deposited under a DOI. Which measures did the authors take to distinguish true structural variants from possible assembly artifacts. This is of particular relevance for the inversion linked to the flesh color gene.

Reply: Thanks for pointing this out. We performed manual inspection of the chromosome-scale scaffolds in the light of Hi-C data after the initial run of 3D-DNA pipeline. One example of three falsely ordered contigs is shown in **Response Fig. 3**, in which we manually repositioned the three contigs by inverting them based on Hi-C interaction intensity.

Response Fig. 3: An example of manual curation of Hi-C scaffolding. The left panel shows the Hi-C heat map of the initial scaffolding results. Hi-C interaction patterns after manual curation are displayed in the right panel. Blue rectangles represent super-scaffolds generated by 3D-DNA and green boxes mark the assembled contigs. Black boxes denote the two sets of contigs which required manual inverting.

We have provided Hi-C contact maps as **Supplementary Fig. 4**. For large SVs, we only retained those localized in a single assembled contig to preclude possible assembly artifacts. For accessions with Hi-C data, we also checked the chromatin interaction signals around the SV breakpoints. To validate the 5.8-Mb inversion event on chromosome 3, we used the Hi-C data from the homozygous line A6-26 (DM haplotype) to map to the PG5068 (Wild/CND haplotype) and PG6245 (DM haplotype), whose initial assembled contig encompasses the inversion region in the revised manuscript (**Response Fig. 4**).

Response Fig. 4: An example of Hi-C validated inversion event using DM as the reference genome. Dot plots depict the pair-wise genome comparison around this region between PG5068,

PG6245 and DM. Hi-C contact heat maps at 25-kb resolution are also shown when mapping Hi-C data of the homozygous line A6-26 (DM haplotype) to accession PG5068 (Wild/CND haplotype) and PG6245 (DM haplotype).

Minor points:

L. 29: is a paradigm shift. That's too optimistically phrased. The recent paper by the Huang group on hybrid potato is indeed impressive. Their approach has the potential to usher in a paradigm shift in potato breeding, but this potential has yet to be realized.

Reply: We have changed "Diploid hybrid breeding on the basis of true seeds is a paradigm shift in potato breeding" to "Recent advances in diploid hybrid breeding based on true seeds have the potential to revolutionize future potato breeding and production".

L. 35: wide diversity: without quantification this is not very meaningful, it may be better to simply say ... accessions from the *Solanum* section *Petota*.

Reply: We have changed "which represent a wide diversity of *Solanum* section *Petota*" to "representative of *Solanum* section *Petota*".

L. 44 controlling a nutritional trait: I'd prefer a more specific phrasing spelling out the trait.

Reply: We have changed "a nutritional trait" to "carotenoid content".

L. 56 facilitate more rapid -> speed up

Reply: We have changed "facilitate more rapid genetic improvement" to "accelerate genetic improvement"

L. 60 empower: that's a complicated way for saying potato reproduce vegetatively by tubers

Reply: For clarity, we have rephrased this sentence to "Furthermore, the impact of the evolution of a clonal reproduction strategy on potato genomes and the evolutionary mechanisms of tuberization are largely unexplored".

L. 60 its: it's not quite clear to what 'it' refers to. Given that reproduction is the *raison d'être* for tuberization, the logic of this sentence looks a bit muddled

Reply: We have rephrased the sentence "The vegetative reproduction strategy of potato is

empowered by tubers, while its impact on potato genomes and evolutionary mechanisms of tuberization are largely unexplored” to “Furthermore, the impact of the evolution of a clonal reproduction strategy on potato genomes and the evolutionary mechanisms of tuberization are largely unexplored”

L. 77 it will be more informative to report coverage instead of leaving the reader to figure out genome size and do the calculations themselves

Reply: We have changed “an average of 24.5 Gb high-fidelity (HiFi) reads” to “an average of 24.5 Gb (~30-fold relative to the estimated haploid potato genome size of ~800 Mb) high-fidelity (HiFi) reads”.

L. 79 introduce the terminology

Reply: To avoid confusion, we have revised this sentence to “raw assembled contigs with heterozygous regions retained and monoploid assembled contigs (MTGs)”.

L. 82 Hi-C: at least spell out the abbreviation, possibly refer to relevant papers on Hi-C scaffolding (Burton et al./Kaplan et al.)

Reply: We have inserted the full name of Hi-C and added these two references in the revised manuscript.

L. 97/98: I’m OK with the conclusion, but the logic is not clear, and I don’t agree with the premise. Seeing a plateau curve ED Fig. 1d is in the eye of the beholder. I don’t see one. It’s probably not necessary to do a full-fledged saturation analysis (whose conceptual basis is also vague without knowing the universe). Some more cautious phrasing would do.

Reply: We agree with the Reviewer and according to the comments of Reviewer #2, we have rephrased this sentence to “suggesting that our panel captures the shared gene content of potato”.

L. 100 soft/shell: terminology in need of introduction

Reply: We have changed “soft-core clusters (5,743, 11.2%), shell clusters (28,471, 55.4%)” to “soft-core clusters (present in 42-44 accessions, 5,743, 11.2%), shell clusters (found in 2-41 individuals, 28,471, 55.4%)”.

L. 103: Interpro protein: technical jargon, better: protein with domains annotated in the Interpro database

Reply: Thanks for the advice. We have changed “InterPro protein domains” to “protein domains annotated in the InterPro database”.

L. 108: Ka/Ks is not well-defined in intraspecific comparisons (see PMID: 19081788). Your panels is a mixture of samples for the same and from different species. Ka/Ks makes only in between species

Reply: We agree with the Reviewer and admit that our accession panel harbors an array of diverse species. Therefore, within each cluster containing genes from the 45 potato accessions, we estimated K_a/K_s between a pair of genes from two randomly selected accessions. This process was independently replicated 50 times for each cluster. We thus added a “pair-wise” before non-synonymous/synonymous substitution ratios (K_a/K_s) in the revised manuscript.

L. 122: Given the importance later in the text, a definition of stolon is appreciated

Reply: We have rephrased the sentence “*Etuberosum* species generate rhizomes, resembling potato stolons, which, however, grow upwards to form new daughter plants without bearing tubers” to “Potato stolons are underground shoots or stems capable of bearing tubers, whereas *Etuberosum* species generate rhizomes, resembling potato stolons, which grow upwards to form new daughter plants”.

L. 126: long reads: be more specific, CLR and HiFi are very different.

Reply: Sorry for the confusion, we have changed “PacBio long reads” to “PacBio continuous long reads”. This was also described in the Manuscript (Lines 137-138).

L. 140: This sentence doesn’t fit into the context. What are the traits? Do you mean to compare single-locus vs. genome-wide trees?

Reply: We have rephrased this sentence to “Phylogenetic topologies based on a single gene or genomic region may disagree with species topologies inferred from whole-genome markers”.

L. 156: extensive interspecific hybridization: I’m not well-read in the potato literature. But I expect this phenomenon may be observed before with different flavors of molecular markers. If so, the relevant papers should be cited

Reply: Thanks for the suggestion and we have added a citation: “Spooner, D. M., Ghislain, M., Simon, R., Jansky, S. H. & Gavrilenko, T. Systematics, diversity, genetics, and evolution of wild and cultivated potatoes”.

L. 162: rapid decay: I think this misses the point. Selfing crops were often domesticated from selfing wild progenitors. So heterozygosity is low in either taxon.

Reply: This sentence has been removed and we have substantially re-written this section to present insights that our pan-genome analyses can offer into hybrid potato breeding.

L. 165: increased heterozygosity: relative to what? To selfers, yes, but that's almost self-evident. Wild vs domesticated? It's not clear to me why domestication per se should increase heterozygosity.

Reply: This sentence has been removed and we have substantially re-written this section to present insights that our pan-genome analyses can offer into hybrid potato breeding.

L. 168 ATG/PTG: define terminology

Reply: We have provided details about ATG and changed PTG to MTG (monoploid assembled contig), as mentioned in the section "**Pan-genome of the *Petota* section**" on Line 87.

L. 174 Can you rule higher homozygosity due to a very recent bottleneck on a population genetic time scale, rather a change of mating system in phylogenetic time?

Reply: Due to the lack of evidence to support this statement, we have determined to remove this description, as also mentioned in the comments of Reviewer #2.

L. 181: I think you refer to *deleterious* variants. The accumulation of genetic variants ultimately traces back to incoming cosmic radiation, on which clonal propagation has no bearing.

Reply: According to the comments of Reviewer #2, we have removed this sentence and generated new results on the construction of a map of large-scale inversions, which has also been arranged into **Fig. 4a**. Please see the revised manuscript (Lines 326-332).

L. 189: SL4.0 technical jargon

Reply: Thanks for pointing this out. We have changed it to "Heinz 1706".

L. 191: This conclusion is odd. In ED Fig. 4, everything looks quite collinear. Your phrasing is vague. A concrete scenarios would be the accumulation of structural variants that dramatically reshuffle genes, but ED Fig. 4 shows nothing of the kind. It needs to be spelled out was you mean by synteny (at which scale?) and by substantial loss.

Reply: To generate the dot plots shown in Extended Data Fig. 2 in the revised manuscript (Extended Data Fig. 4 in the original manuscript), we used a whole-genome alignment algorithm embedded in the mummer software to compute the synteny regions between the 44 potato accessions and DM, the reference genome. Therefore, Extended Data Fig. 2 simply depicts the genome-wide synteny between these potato genomes. The dot plots indeed look collinear, because the size of the point did not represent the actual proportion of the genome. One point may just denote an aligned segment of tens of kb.

The syntenic proportion reported in the main text (28.0% in potato and 87.0% in tomato) was calculated by performing whole-genome alignment, using the DM genome as the reference by Progressive Cactus⁷ (please see the **Methods** for more details). We have also removed the “substantial” in the main text, since it is quite hard to quantify the degree of “loss”.

L. 193: Founder is a term from pedigree breeding. Is it well-defined for landraces?

Reply: We have removed this description in the revised manuscript.

L. 195: The sentence stands without the relative clause, which I think makes the implicit assumption that you are talking about *deleterious* variants. One does not purge something beneficial.

Reply: We have removed this description in the revised manuscript.

L. 204-207: I'd like to see some more technical validation of the inversions, e.g. by showing Hi-C contact matrices or genetic maps. If the inversions are contained in single HiFi contigs, that's also strong support their validity.

Reply: We have added local Hi-C contact maps of two accessions showing the contact pattern of the presence or absence of the inversion in **Fig. 4b**. We checked the genomic coordinates of the inversion and found that this inversion resided in a single assembled contig in the PG5068 genome. These results indicate the validity of this inversion.

L. 211: left The genome knows no left or right. Distal and proximal are better terms.

Reply: We have rephrased this sentence to "This gene was located ~1.5-2 kb proximal to the breakpoint of the 5.8-Mb inversion".

L. 217-219: *is the causative variant* This conclusion is too strong. This can only be proven by re-creating the inversion by gene editing. Rephrase in a more cautious way.

Reply: We agree with the Reviewer and have removed corresponding description of "causative" variation. We also rephrased the corresponding description in the abstract.

L. 222 hampering: It's conceivable that inversions create beneficial super-genes that actually boost crop improvement.

Reply: We have removed "hampering crop genetic improvements" to avoid confusion.

L. 228: I don't think "purposefully" is needed here. The purpose (hybrid breeding) is understood. Maybe rephrase: select parental lines with optimal nutritional profiles.

Reply: We have rephrased this sentence to "breeders could now select appropriate donor or acceptor lines for backcrossing".

L. 232: The logic of this sentence is a bit muddy. How about: Clonal propagation gave rise to tuber-borne diseases against which potato has evolved a repertoire of novel resistance genes.

Reply: Thanks for the suggestion. We have rephrased this sentence to "Clonal propagation gave rise to the emergence of tuber-borne diseases against which potato has possibly evolved an expanded repertoire of resistance genes".

L. 240: The switch to tomato needs a better motivation. At first, I had the impression that you were almost literally comparing apples and oranges.

Reply: The reason that we chose the tomato dataset to examine our approach is that tomato is the only species, sufficiently close to potato, which has a third-generation sequencing-based RenSeq-derived NLR dataset. NLRs predicted throughout the RenSeq approach have widely been considered as the highest quality to date. To be clearer, we have rephrased this sentence to "To mitigate this, we developed a "NLR local annotation" pipeline and benchmarked it with a tomato NLR dataset, based on resistance gene enrichment sequencing (RenSeq), resulting in comparable numbers of NLRs".

L. 250: monoploid: see my comment above about QC on how well "haploidization" in hifiasm works

Reply: Please see the above responses to your second concern. We also removed "PTG" in this sentence to be more straightforward.

L. 262-267: This is a short paragraph on a complex analysis comparing two species unrelated to

potato. An obvious technical concern if the Ipomoeae assemblies are good enough to support this analysis

Reply: The genomes of *I. trifida* and *I. triloba* were assembled using mostly Illumina short reads, whereas the genome of *I. nil* was constructed using PacBio CLR data^{8,9}. In this section, we only retrieved *NLR* genes from the protein-coding gene prediction datasets of the three *Ipomoeae* species and compared their copy numbers. Since assemblies of genomes built using Illumina short reads can also achieve a high coverage of genic regions, we believe this comparison will not be biased by the different assembly qualities. To validate this, we compared the number of NLRs in different classes between two versions of the potato reference genome DM (short read-based v4.03 and long read-based v6.1) and did not observe notable difference between them (**Response Fig. 5**).

Response Fig. 5: Comparison of NLR copy numbers between two versions of the potato DM reference genomes, built using Illumina short reads (v4.03) and Nanopore long reads (v6.1).

L. 284-285: “whole genome alignment considering evolutionary distance” This is vague. It leaves the reader in the dark about what was actually done. The Methods are more informative, but a better summary is needed here.

Reply: We have rephrased this sentence to “We identified 149,663 potato-specific CNSs (6.9 Mb) by computing conservation scores, based on whole-genome alignment, using genome sequences of 45 potatoes (including DM), 24 tomatoes, and two *Etuberosum* species”.

L. 349: To my understanding, “it has not eluded us” is a tongue-in-check way of introducing some obvious fact. It does not fit here.

Reply: We agree with the Reviewer and have removed this phrase.

L. 361: hybrid designation: I’m not sure what this means.

Reply: Sorry for the confusion, we have removed this sentence.

L. 495: Instead of saying “standard” refer to the relevant protocol.

Reply: We have revised this sentence and added a citation (Belton, J.M. *et al.* Hi-C: a comprehensive technique to capture the conformation of genomes. *Methods* 58, 268-76 (2012).). Please see the revised manuscript (Line 515).

Referee #2 (Remarks to the Author):

In this manuscript entitled "Genome evolution and diversity of wild and cultivated potatoes", Tang and colleagues assemble and analyze high-quality genomes of 44 diploid potatoes, including both wild and cultivated accessions, with the aim of improving our understanding of the potato genome and tuber evolution. Assembling and analyzing this number of diploid, often highly heterozygous genomes is a technical feat. Many of the genomic analyses presented are largely descriptive and essentially provide a catalog of genomic variation among potato accessions. For instance, the study describes a potato pan-genome, identifies structural variants and potato-specific conserved non-coding regions, and describes the distribution of homozygous and heterozygous genome segments in the sequenced potato accessions, as well as the types and numbers of resistance genes and the distribution of syntenic segments. The manuscript also presents analyses of the phylogenetic relationships among potato and related species, as well as functional work, including the identification of a likely tuber initiation gene using CRISPR-Cas9-based genome editing. The manuscript thus presents a substantial body of work aiming to document genomic variation and provide insights into the genetic basis of tuber formation in potatoes. The manuscript is mostly clearly written and the figures are clear.

Although genomic diversity in tuber-bearing *Solanum* has recently been described (e.g. Hardigan et al. 2017. PNAS), this manuscript improves on earlier studies by basing its analyses on highly contiguous genome assemblies. The study relies on the latest methods for long-read sequencing and assembly, and the resulting genome assemblies seem to be overall of high quality. That being said, the evolutionary genomic analyses undertaken are not very novel, and do not yield major novel biological insights. So, while the genomic resources generated will be very valuable for further work on this important crop system and also provides opportunities for evolutionary genomic analyses, the results presented here do not stand out in terms of originality and significance. While genomic analyses seem well executed for the most part, there is frequently a disconnect between the conclusions drawn and the conclusions that would be possible to support based on the conducted analyses. This is particularly the case for the analyses of genomic patterns, where the conclusions drawn in the manuscript are often not supported by the analyses or the data, but also occurs in connection with some of the functional analyses. Furthermore, some analyses are not described in sufficient detail, resulting in poor reproducibility, and statistical tests are sometimes not correctly described.

In sum, while it is certainly of interest to crop breeders to have access to these genome assemblies and the catalogue of genomic variation, I am not convinced that the main conclusions are supported, nor that the study will be of sufficient interest for a very broad and general readership. Below, I outline both major and minor concerns in more detail.

Reply: We thank the Reviewer for these comments. In the revised manuscript, we have performed additional analyses and generated extra datasets to support our conclusions regarding the genome evolution. According to the Reviewer's advice, we have rephrased some of the conclusions that could not be well supported by current analyses owing to the unavailability of data for an appropriate control group. However, the discovery of genomic heterozygosity patterns, loss of synteny compared with tomato, and the prevalent large-scale inversions, indeed offers critical implications for hybrid potato breeding. Thus, we have also substantially re-written and integrated the two original sections "**Genomic architecture of heterozygous diploid potato**" and "**Extensive divergence in *Petota***" into "**Pan-genome-guided hybrid potato breeding**", with the aim to facilitate hybrid potato breeding, using our pan-genome analyses. We have also added more details on analytic methods and statistical approaches. We believe that these will sufficiently address the Reviewer's concerns without sacrificing the global impact and the broad reader interest of this manuscript.

Major comments

1. Conclusions on hybridization are not supported by analyses

The phylogenetic reconstructions done to clarify the position of *Etuberosum* are nicely done, and I appreciate that the authors used a combination of super-matrix analyses and coalescent-based analyses to reconstruct a species tree. However, the authors then go on to conduct analyses in 100 kb non-overlapping windows, and use the widespread gene tree-species tree discordance to argue for a "complex history of potato evolution including extensive interspecific hybridization" (Lines 151-154). There are several problems with this interpretation. The first one is methodological – based on the methods it seems that for this analysis, all non-coding regions were kept, and there is no mention of filtering out repetitive non-coding regions. This is problematic, as erroneous SNP calls in repetitive regions are common and can result in gene tree discordance even in the absence of interspecific hybridization (or for that matter, incomplete lineage sorting). If repeats were not already filtered, the analysis should be redone after doing so.

Reply: We appreciate the Reviewer for pointing out this. We have reperformed this analysis using the alignment among these genomes that have been hard masked for repetitive sequences and obtained similar results. The set of window-based trees showed in **Fig. 1b** have now been updated accordingly.

However, the second problem is possibly inherent to the system being studied. When analyzing genomic data from a set of closely related species, it is not necessarily unexpected to observe gene tree discordance, and the results that are presented do not allow the authors to draw conclusions on the causes of the discordance. To be able to draw such conclusions much more rigorous analyses would be needed, taking into account that gene trees often do not have the same topology as the

species tree when internodal branches are short relative to the effective population size (as shown by Pamilo and Nei 1988). As the authors do not present any estimates of divergence time (e.g. based on synonymous divergence), nor provide information on intraspecific polymorphism at neutrally evolving sites (which is related to the effective population size), it is difficult to develop an intuition for how rampant incomplete lineage sorting would be expected to be in this case. However, it is clear that based on the presented analyses, it is not possible to draw any firm conclusions on the frequency of interspecific hybridization.

Reply: Thank you for the comments. We have applied more analyses to assess the impact of ILS and/or hybridization on the discordance between gene trees and the species tree. The lineage sorting processes might be incomplete among evolutionarily closely-related species, considering that *Etuberosum* diverged with the common ancestor of *Lycopersicon* and *Petota* at 8.30 million years ago (MYA, 95% highest posterior: 7.9-8.8 MYA). To estimate the contribution of ILS to the incongruence, we applied the approach described in Scally *et al.*¹⁰ and observed 21.6 – 24.7% of the potato genome exhibiting ILS, under the estimated topology ((*Lycopersicon*, *Petota*), *Etuberosum*), *S. melongena*).

We also computed the theta parameter, which reflects the effective population size, by dividing the mutation units for each internal branch by coalescent units¹¹ (**Response Fig. 6a**). Moreover, we used DendroPy¹² to simulate 20,000 trees with ILS using six potato species (*S. tuberosum* Group *stenotomum*, *S. candolleianum*, *S. lignicaule*, *S. chacoense*, *S. cajamarquense*, and *S. bulbocastanum*), setting *S. melongena* as the outgroup (**Response Fig. 6b**). The positive correlation ($R^2 = 0.81$, P -value $< 2.2e-16$) between the occurrence frequencies of these topologies and those from the 3,971 single-copy gene trees also supported the presence of ILS among these species.

Response Fig. 6: The impact of ILS on the tree discordance. **a**, Phylogenetic tree with branches being colored by the inferred population mutation parameter theta, which reflects the population polymorphism, by dividing the mutation units for each internal branch by coalescent units¹¹. The grey color denotes branches where theta is unable to be computed due to lack of data. **b**, Simulation of 20,000 gene trees with ILS by DendroPy indicates a positive correlation between the observed and simulated gene-tree topologies from all possible four-species groups among the six potato species.

Using the D statistics¹³, we detected gene flow between the species in *Petota* and *Etuberosum* sections ($D = 18.9\%$, $Z = 30.6$), and f_4 -ratio statistics¹⁴ showed that 8.4% of the potato genome displayed admixture between *Petota* and *Etuberosum* (**Response Fig. 7**).

Response Fig. 7: ABBA-BABA analyses of gene flow between *Petota* and *Etuberosum* species. Significant introgression events are detected between *Petota* and *Etuberosum*. *S. melongena* is set as an outgroup.

To estimate the prevalence of gene flow within *Petota*, we also performed the D statistics among the 25 *Petota* accessions and detected 1,069 out of 2,300 triplets showing significant signals ($Z > 3$) of gene flow (**Response Fig. 8a**). Moreover, we asked whether there was gene flow between the landraces and wild species. We calculated D -statistics in the form of $D(\text{landrace, landrace}; \text{wild, outgroup})$, which is expected to be zero if gene flow was absent. We observed compelling gene flow signals related to *S. ajanhuiri* (PG6002, **Response Fig. 8b**), and the signals were the strongest when testing with *S. boliviense* (PG5076) among all the wild species (**Response Fig. 8c**), suggesting that gene flow has occurred between the landrace *S. ajanhuiri* and its sympatric wild species, *S. boliviense* in Bolivian. These results have been added in the revised manuscript.

Response Fig. 8: Frequent inter-specific hybridization within *Petota*. **a**, Heat map of the most

significant D scores observed between two given potato accessions (P2 and P3) across all possible individuals in P1 species. D scores and $\log_{10}(p)$ values are shown in different color schemes. Tomato, *Etuberosum* species, *S. americanum* and *S. melongena* are used as outgroups. **b**, Quantile-quantile plot comparing Z scores from all combinations of D (landrace, landrace; wild, outgroup) to those expected under a normal distribution. The asymmetric signal is driven by *S. ajanhuiri* (PG6002), which consistently shows closer affinity with wild species than do other landrace accessions. **c**, The Z scores for D (*S. ajanhuiri*, landrace; wild, outgroup) divided by the wild accessions at the P3 position of the D -statistics along the x-axis.

2. Conclusions on reproductive mode are not supported by analyses

The authors investigated substantial variation in the distribution of homozygous and heterozygous segments in the sequenced genomes and drew the conclusion that "These results suggest that the reproduction strategy incorporating both tubers and seeds of potato in nature resulted in the heterogeneous genomic architecture" (lines 172-174).

It is unclear how the authors can assess the impact of the complex reproduction strategy here, as there is no control group with a different reproduction strategy included in the analyses, and it is also not clear why the observed variation in the distribution of heterozygous segments is seen as remarkable.

Furthermore, the authors state (on lines 169-172) that "We also identified eight near-homozygous chromosomes in *S. morelliforme* (PG1011, Chr1, 2, 4, 6, 7, 10, 11, and 12) (Extended Data Fig. 3a), which is a self-incompatible species suggesting a recent event of conversion from self-compatibility to self-incompatibility in this accession."

It is not uncommon that even self-incompatible species can be subject to inbreeding (e.g. biparental inbreeding), which could result in homozygous segments. Therefore, observing a self-incompatible species with high homozygosity cannot be used to conclude that it was previously self-compatible. Given the evolutionary rarity of transitions from self-compatibility to self-incompatibility, the authors' claim is extraordinary, and much stronger empirical evidence (ideally on the detailed genetic basis of potentially re-evolved self-incompatibility) would be required to draw such a far-reaching conclusion.

Given the lack of a control group and the insufficient support for the conclusions drawn, I would recommend to exclude the analyses of heterozygosity from the main text altogether, or to improve the analyses by including a control group.

Reply: We agree with the Reviewer's comments. Due to the lack of an appropriate and feasible control group with similar sequencing and assembly strategies (PacBio HiFi-based assemblies with heterozygous regions mostly resolved), we have removed the descriptions regarding reproduction strategies and the conclusion relevant to *S. morelliforme*. The analyses of length and distribution of heterozygous and homozygous regions of diploid potatoes was retained and we

have also added additional analyses on hemizygous genes. The important implications of our analyses to hybrid potato breeding have been presented. Please see the revised manuscript in Lines 288 - 305.

3. Lack of evolutionary context for analyses of synteny

The authors use multi-way alignments to compare synteny among sequenced genome of cultivated tomato vs cultivated potatoes. They document " a substantial loss of synteny in cultivated potatoes, probably due to the elevated level of genetic divergence by tuber propagation and relatively high founder diversity¹⁵." First, this analysis demonstrates the utility of a control group – it is very nice that the authors compared their results to those for cultivated tomatoes. However, assembly errors particularly in heterozygous genomes could affect inference of synteny, so it would be important to document that the assemblies of the contrasted groups are similar in quality. Assuming that the results are robust to technical artefacts, in order to fully interpret these results, more information on the level of divergence among the compared tomato accessions relative to the divergence among the potato accessions is still required. The tomato genomes analyzed are not included in any phylogeny shown, so one cannot rule out higher synteny just due to more recent divergence among the tomato accessions analysed. The authors should provide more evolutionary context to facilitate interpretation of these results.

Reply: We thank the Reviewer for pointing out this. We have applied the latest assembly methodology to construct the heterozygous genomes of the 44 potato accessions. The monoploid assemblies for these genomes were then extracted to perform the synteny analysis. The tomato genomes investigated in this study were all built using the third-generation sequencing technique (PacBio CLR and Nanopore) and are all assembled into 12 chromosomes. These results suggest relatively high qualities of the potato and tomato genomes investigated here.

We also inferred the phylogeny among the 73 accessions used in this study (45 potatoes, 24 tomatoes, two *Etuberosum*, one *S. americanum* and one *S. melongena*) and estimated the divergence time (**Response Fig. 9**). We observed that wild and cultivated tomato occurred more recently than potato. Given the difference of divergence time, we have removed the corresponding conclusion on tuber propagation and retained the results of genome-wide synteny in terms of both sequence and gene levels to provide a critical implication for hybrid potato breeding.

Response Fig. 9: The estimated divergence time of the 73 accessions used in this study. The numbers denote the estimated divergence time (million years ago) between key species/clades.

4. Causal nature of inversion not supported

The authors identify a 5.8 Mb inversion close to the Y locus that determines tuber flesh colour. After comparing haplotypes and expression data for accessions with white and yellow tuber colour, the authors conclude that "the 5.8-Mb inversion is the causative variant that provides a new promoter to activate the downstream of BCH gene". This statement is not strictly supported, as genetic modifications would be needed to test this hypothesis. For now, it is not possible to rule out sequence differences in the 1.5-2 kb next to the BCH gene but outside of the inversion. For that reason, this conclusion should be rephrased.

Reply: We agree with the Reviewer and have removed the sentence "the 5.8-Mb inversion is the causative variant that provides a new promoter to activate the downstream of *BCH* gene".

Notwithstanding, we have added new results on large-scale inversions and generate useful insights into hybrid potato breeding. Large inversions have been reported to suppress recombination by reducing crossing-over^{15,16}, resulting in severe linkage drag when conducting backcross breeding. To avoid this, it is necessary to select donor lines without inverted fragments harboring target genes. Therefore, we constructed a map of large-scale inversions, which is rather challenging to be resolved using resequencing data and methodology¹⁷. The results regarding the 5.8-Mb inversion were considered as an example to indicate the importance of avoiding large inversions in backcross breeding (Lines 341-346). These results have been added in the revised manuscript.

Comments on statistical analysis/description of methodology

5. Data and analyses underlying selection of accessions is not sufficiently described. The authors state that their choice of accessions was based on phylogenetic relationships of 432 accessions (lines 71-73) and refer to Supplementary Table 1 and Supplementary Figure 1. The methods states that all these accessions were sequenced using Illumina technology, but Supplementary Table 1 does not list Illumina data for all of those accessions. There is no description of how the data were processed to identify SNPs and how those SNPs were filtered, how many SNPs were analysed, how the phylogeny was generated (no software or sequence evolution model is mentioned), and bootstrap values are missing so there is no information on the support of different branches. Furthermore, it is not possible to connect Supplementary figure 1 to Supplementary table 1 as there are no labels on the taxa in the phylogeny. It is also not clear from the Reporting Summary whether the Illumina data is made available through NCBI. In short, this part of the analysis is not reproducible based on the information provided.

Reply: Among the 432 accessions, 33 are from Hardigan *et al.*¹⁷ and 201 are from Li *et al.*¹⁸ Illumina resequencing data of the remaining 198 out of the 432 accessions are still being analyzed by another lab; therefore, we regret that the raw sequencing data cannot be made publicly accessible currently, and we can only provide the SNP information of these 198 lines. However,

we consider that the SNP information, and the previously reported Illumina sequences, enable the reproducibility of this analysis. Detailed descriptions of computational methods used to process these Illumina data and to identify genetic variants, as well as that of phylogeny related analyses, have been integrated into the **Methods** section (Lines 500-504). We have also added bootstrap values and names of accessions in **Supplementary Fig. 1**. The genotype information of the 432 accessions has been deposited in our Pan-Potato Database (<http://solomics.agis.org.cn/potato/ftp/>) and has been listed in the **Reporting Summary**.

6. In the pan-genome analysis, the authors show that Ka/Ks values are lower for core genes than for accessory genes (called shell and accession-specific) and that expression was higher for core genes. This is a result that is in line with expectations if core genes are under stronger functional constraint, perhaps in part due to their higher expression levels and due to the inclusion of "housekeeping" genes in this category. To fully report how they arrived at this result, however, the authors would need to provide more information on how Ka/Ks estimation was done. Specifically, the KaKs_Calculator run by ParaAT can estimate Ka/Ks using a number of different methods, and here one would need some more information on which one was used. It is also not clear what is meant by "50 different combinations of Ka/Ks within each cluster were computed".

Reply: We are sorry about the lack of clarity. The default parameter of KaKs_Calculator was set to estimate the K_a/K_s values, which means the K_a/K_s was the average of the output from 15 available algorithms comprising seven original approximate methods (NG, LWL, MLWL, LPB, MLPB, YN and MYN), seven gamma-series methods (γ -NG, γ -LWL, γ -MLWL, γ -LPB, γ -MLPB, γ -YN and γ -MYN) and one maximum likelihood method (GY)¹⁹. We have added these technical details in the **Methods** section (Lines 597-601).

For clarity, we have changed "50 different combinations of K_a/K_s within each cluster were computed" to "Within each cluster, K_a/K_s between gene pairs from 50 randomly chosen combinations of two accessions were estimated". Please see the revised manuscript (Lines 603-604).

7. Statistical treatment or notation unclear

With respect to the pan-genome analysis, it is unclear how the statistical comparison of gene clusters were done, specifically regarding the data presented in Extended data figure 1 f and g. The letter labels above each category in these graphs implies that a post-hoc test has been done after a significant Kruskal-Wallis test but the figure legend states "Multiple comparison was performed using Kruskal-Wallis test with $\alpha = 0.001$ ". As the Kruskal-Wallis test does not compare groups in a pairwise manner, more information is needed to support the figure. What specific post-hoc multiple comparison test was done, with what settings? The text in methods (lines 565-567) is not sufficient.

Reply: Multiple comparisons were performed using the Fisher's least significant difference implemented in the “kruskal” function in the R package “agricolae”. The level of significance used in the *post hoc* test was 0.001. We have also modified the corresponding description in the figure legends and the **Methods** (Lines 607-609).

Inconsistent regarding statistical notation is also used regarding reported correlations (line 82-85, Extended Data Fig. 1c), please use *r* or *R* consistently for Pearson's correlation coefficient.

Reply: We have changed the “*R*” in **Extended Data Fig. 1c** to “*R*²”.

7. How the phylogenetic analyses were done is not described in sufficient detail in the methods. For instance, it is stated that ” To build a super-matrix species tree, amino-acid sequences of representative annotated gene models were first extracted from genomes of 44 PTGs, DM v6.112, 24 tomato accessions (see Supplementary Table 1 for more details)26,42,74,75, two *Etuberosum* species and eggplant81.”

Supplementary Table 1 contains more information on the accessions but not on the ”representative annotated gene models” – how were they selected, and how many single copy clusters were included in the IQ-TREE analysis?

Reply: We have revised the relevant description in the **Methods** section (Lines 639-647). The representative annotated gene models were the longest transcripts from a given predicted gene (Line 640). We have also added the number of single-copy gene clusters into the **Methods** (Line 645).

Minor comments on data availability and presentation

8. Data availability unclear

It is not clear whether all data, including Illumina whole-genome resequencing data used for accession choice, has been uploaded to an appropriate repository such as NCBI. The Data part of the Reporting Summary statement only lists PacBio data, transcriptome data, and Hi-C data as having been deposited there, and a reference to a custom database with unknown access terms (the database is not currently accessible) is provided for ”genome assemblies, annotation, sequence variation, gene expression”. Note however that the Methods states that all sequence data has been uploaded to NCBI – this is currently unclear.

Reply: As mentioned earlier in the Reviewer's concern, a total of 198 out of the 432 accessions were not generated by us and are still being analyzed by another lab; therefore, we regret that the raw sequencing data cannot be made publicly accessible currently, but we can provide the genotype information of all the 432 lines, which have been available in our Pan-Potato Database

(<http://solomics.agis.org.cn/potato/ftp/>) and it has now been accessible. If the website is not available for any reason, please try this URL instead (<http://218.17.88.60/potato/ftp/>). Illumina data for the remaining 234 accessions were downloaded from previous research^{17,18}. We have added more details into the **Data and code availability** section. This information is sufficient for sample selection.

9. Lines 95-97 ” Pan-genome size increased when incorporating more genomes and reached a plateau when n was close to 40 (Extended Data Fig. 1d), suggesting that our collection represents the broad biodiversity of potato.”

Rephrase to ”our collection captures the shared gene content of potato” as not all aspects of genetic diversity or biodiversity are captured by variation in gene content.

Reply: Thanks for the suggestion. We have rephrased this sentence to “our panel captures the shared gene content of potato”.

10. In the abstract on (line 31-32) the following sentence should be rephrased: ” However, it remains under-investigated on the genome evolution and diversity of wild and cultivated potato, limiting the utilization of rich variation in the potato gene pool.”

Reply: Thanks for the suggestion. We have rephrased this sentence to “Currently, the genome evolution and diversity of wild and cultivated landrace potatoes remains relatively unexplored, limiting the utilization of their diversity in potato breeding”.

Referee #3 (Remarks to the Author):

A. Summary of the key results

Using new sequencing technologies and de novo genome assembly software, authors delivered pan-genome of heterozygous potatoes covering diploid and wild species. The study presents new insight into the genetics and evolution of tuber formation in plant species by providing comprehensive genome and transcriptomes of Petota as well as a Lycopersicon and Etuberosum accessions. It also sheds some light on structural variation affecting the expression of genes encoding for yellow flesh, which is an important nutritional trait and important target for genetic improvement. These genomic resources improve our understanding of interspecific variation and genome evolution in potatoes. They also provide the foundation for genomics-informed potato breeding and genetic resources management.

Reply: Thanks for the comments!

B. Originality and significance: if not novel, please include reference

The study extends our knowledge of potato evolution by analysing genome of accessions from Solanum species belonging to clades 1-4. The results are of immense interests to potato breeders and gene bank curators to guide efficient breeding and genetic resources management strategies.

Reply: Thanks for the comments!

C. Data & methodology: validity of approach, quality of data, quality of presentation

All data were thoroughly reviewed and no flaw was detected. However, since I am not an expert in bioinformatics and taxonomy, some details may have escaped me.

Reply: Thanks for the comments! Based on comments from Reviewer #1 and Reviewer #2, we have added additional technical details and supporting materials in the revised manuscript.

D. Appropriate use of statistics and treatment of uncertainties

All data were thoroughly reviewed and no flaw was detected. However, since I am not an expert in bioinformatics and taxonomy, some details may have escaped me.

Reply: Thanks for the comments! According to comments from Reviewer #1 and Reviewer #2, we have provided more detailed information about statistic tests performed in the revised manuscript to avoid potential confusion.

E. Conclusions: robustness, validity, reliability

The panel of germplasm used included tuberizing and non-tuberizing species from the Solanaceae family, as well as outbreeders and inbreeders from a wide range of geographical distribution. Most of the discussion is centered around the impact of tuberization on potato genomes and evolutionary mechanisms, whereas a possible role of ecological distribution and interspecific hybridizations or introgressions was not as much covered. Overall, conclusions are valid and solidly founded.

Reply: We believe that this is an exploration of the potato pan-genome specifically for wild species. Future studies, coupled with the access to phased tetraploid potato assemblies, will gain entry to examine introgression patterns. Regarding discussion of ecological distribution, we have added a relevant discussion on Lines 362-376 in the revised manuscript.

F. Suggested improvements: experiments, data for possible revision

A minor revision is suggested to include discussion on possible role of species geographical distribution

and breeding behaviour on potato genome evolution in the context of the present study.

Reply: We have added a corresponding discussion in the revised manuscript (Lines 362-366) as follow: Geographical isolation between the North and South American continent may contribute to the species from clades 1+2 being the sister lineage to wild species in clades 3+4 and other landraces. Previous reports indicated that diploid cultivated potatoes were domesticated from wild potatoes from clade 4^{17,18}. In this study, we also found that *S. candolleianum* is sister to cultivated potatoes further supporting this species as the immediate progenitor of cultivated potato. Future studies, coupled with the access to phased tetraploid potato assemblies, will allow the examination of introgression patterns from wild species, as introgression breeding was mainly conducted in these tetraploid cultivars.

We have also added discussion on the implication of our work to the utilization of wild germplasm: Considering the endosperm balance number (EBN), a hypothetical unified prediction concept of crossability, between the majority of the wild species (17 out of 24) investigated here and the cultivated potatoes is the same (2EBN), the pan-genome will motivate attempts for the introgression of favorable traits from these wild species to breed better inbred lines.

G. References: appropriate credit to previous work?

Most relevant references were included.

Reply: Thank you for the comments.

H. Clarity and context: lucidity of abstract/summary, appropriateness of abstract, introduction and conclusions

The quality of the manuscript is excellent.

Reply: Thank you for the positive feedback.

References

- 1 Cheng, H., Concepcion, G. T., Feng, X., Zhang, H. & Li, H. Haplotype-resolved *de novo* assembly using phased assembly graphs with hifiasm. *Nat. Methods* **18**, 170-175 (2021).
- 2 Potato Genome Sequencing, C. *et al.* Genome sequence and analysis of the tuber crop potato. *Nature* **475**, 189-195 (2011).
- 3 Leisner, C. P. *et al.* Genome sequence of M6, a diploid inbred clone of the high glycoalkaloid-producing tuber-bearing potato species *Solanum chacoense*, reveals residual heterozygosity. *Plant J.* **94**, 562-570 (2018).
- 4 van Lieshout, N. *et al.* Solyntus, the new highly contiguous reference genome for potato (*Solanum tuberosum*). *G3 (Bethesda)* **10**, 3489-3495 (2020).
- 5 Zhou, Q. *et al.* Haplotype-resolved genome analyses of a heterozygous diploid potato. *Nat. Genet.* **52**, 1018-1023 (2020).
- 6 Mapleson, D., Garcia Accinelli, G., Kettleborough, G., Wright, J. & Clavijo, B. J. KAT: a K-mer analysis toolkit to quality control NGS datasets and genome assemblies. *Bioinformatics* **33**, 574-576 (2017).
- 7 Armstrong, J. *et al.* Progressive Cactus is a multiple-genome aligner for the thousand-genome era. *Nature* **587**, 246-251 (2020).
- 8 Hoshino, A. *et al.* Genome sequence and analysis of the Japanese morning glory *Ipomoea nil*. *Nat. Commun.* **7**, 13295 (2016).
- 9 Isobe, S., Shirasawa, K. & Hiraoka, H. Current status in whole genome sequencing and analysis of *Ipomoea* spp. *Plant Cell Rep.* **38**, 1365-1371 (2019).
- 10 Scally, A. *et al.* Insights into hominid evolution from the gorilla genome sequence. *Nature* **483**, 169-175 (2012).
- 11 Cai, L. *et al.* The perfect storm: gene tree estimation error, incomplete lineage sorting, and ancient gene flow explain the most recalcitrant ancient angiosperm clade, malpighiales. *Syst. Biol.* **70**, 491-507 (2021).
- 12 Sukumaran, J. & Holder, M. T. DendroPy: a Python library for phylogenetic computing. *Bioinformatics* **26**, 1569-1571 (2010).
- 13 Green & Richard, E. A Draft Sequence of the Neandertal Genome. *Science* **328**, 710-722 (2010).
- 14 Patterson, N. *et al.* Ancient Admixture in Human History. *Genetics* **192**, 1065 (2012).
- 15 Wellenreuther, M. & Bernatchez, L. Eco-evolutionary genomics of chromosomal inversions. *Trends Ecol. Evol.* **33**, 427-440 (2018).
- 16 Huang, K. & Rieseberg, L. H. Frequency, origins, and evolutionary role of chromosomal inversions in plants. *Front. Plant Sci.* **11**, 296 (2020).
- 17 Hardigan, M. A. *et al.* Genome diversity of tuber-bearing *Solanum* uncovers complex evolutionary history and targets of domestication in the cultivated potato. *Proc. Natl. Acad. Sci. U.S.A.* **114**, E9999-E10008 (2017).
- 18 Li, Y. *et al.* Genomic analyses yield markers for identifying agronomically important genes in potato. *Mol. Plant* **11**, 473-484 (2018).
- 19 Wang, D., Zhang, Y., Zhang, Z., Zhu, J. & Yu, J. KaKs_Calculator 2.0: a toolkit incorporating gamma-series methods and sliding window strategies. *Genomics Proteomics Bioinformatics* **8**, 77-80 (2010).

Reviewer Reports on the First Revision:

Referees' comments:

Referee #1 (Remarks to the Author):

The authors have crafted a well-thought-out response letter. The description and documentation of the assembly process are now of sufficient depth. The support of the 5.8 Mb inversion by Hi-C data is unequivocal. The omission of claims concerning self-(in)compatibility and the new analyses of inter-species gene flow have greatly improved the relevant sections of the manuscript. I am pleased to see the meticulous effort the authors took in addressing the referee's concerns. I recommend acceptance.

Referee #2 (Remarks to the Author):

In my original review of the study by Tang and colleagues, I was very impressed by the large number of high-quality genome assemblies generated, as well as by the functional work presented, but found room for improvement regarding the evolutionary genomic analyses. I was very happy to see this revised version, where the authors have substantially revised the text and improved the presentation. I was especially satisfied to see the additional analyses conducted to investigate the evidence for gene flow vs incomplete lineage sorting in more detail. I find that the manuscript is now greatly improved and that the conclusions presented are supported by the data and analyses. I believe that the changes in the text and the structure of the manuscript will also make the manuscript more suitable for a broad and general audience.

I still have concerns regarding data availability. In particular, as the authors outline in their response, they are not making all Illumina data analyzed publicly available (Illumina data for 198 of 432 accessions are not released). While I sympathize with the reasons the authors give for this, providing the sequence data that the results are based on is a standard requirement for publishing genomics papers.

Apart from this major concern, I only have a handful of minor concerns that mainly concern phrasing and statistics:

lines 147-150 "We also estimated that *Etuberosum* diverged with the common ancestor of *Lycopersicon* and *Petota* at 8.30 million years ago (MYA, 95% highest posterior: 7.9-8.8 MYA) (Supplementary Fig. 6)."

Please rephrase "highest posterior" to "highest posterior density interval", and "diverged with" to "diverged from"

lines 307-315 Still not providing evolutionary context for comparison of potatoes and tomatoes, nor establishing that genomes are of comparable quality.

line 502 Please rephrase "fourfold degenerated" to "fourfold degenerate"

line 714-715 "The correlations were computed using the linear regression function "lm()" in R."
Linear regression is not correlation, please check the statistics and redo analyses with cor.test() if required, alternatively change the methods text.

lines 704-705 "To evaluate the theta value for internal branch, which reflects the level of effective population size, and thus the level of ILS, we divided the mutation units by coalescent units"
The expected degree of ILS is not only proportional to the N_e of the internal branch but also the length of that branch (i.e. timing of successive splits). Please correct this statement.

lines 871-882 on Data and code availability. All short-read data should also be deposited in a regular repository for sequence data such as NCBI.

Author Rebuttals to First Revision:

We greatly appreciate the Reviewers' comments to improve our manuscript. Below we provide a point-by-point response to the reviewers' comments and indicate how we have modified the manuscript. In addition, all revisions regarding the reviewers' concerns have been highlighted in **yellow** background in the revised manuscript.

Referee #1 (Remarks to the Author):

The authors have crafted a well-thought-out response letter. The description and documentation of the assembly process are now of sufficient depth. The support of the 5.8 Mb inversion by Hi-C data is unequivocal. The omission of claims concerning self-(in)compatibility and the new analyses of inter-species gene flow have greatly improved the relevant sections of the manuscript. I am pleased to see the meticulous effort the authors took in addressing the referee's concerns. I recommend acceptance.

Reply: Thanks for the comments!

Referee #2 (Remarks to the Author):

In my original review of the study by Tang and colleagues, I was very impressed by the large number of high-quality genome assemblies generated, as well as by the functional work presented, but found room for improvement regarding the evolutionary genomic analyses. I was very happy to see this revised version, where the authors have substantially revised the text and improved the presentation. I was especially satisfied to see the additional analyses conducted to investigate the evidence for gene flow vs incomplete lineage sorting in more detail. I find that the manuscript is now greatly improved and that the conclusions presented are supported by the data and analyses. I believe that the changes in the text and the structure of the manuscript will also make the manuscript more suitable for a broad and general audience.

Reply: Thank you very much. Your earlier comments helped us to conduct additional analyses and to re-structure the manuscript.

I still have concerns regarding data availability. In particular, as the authors outline in their response, they are not making all Illumina data analyzed publicly available (Illumina data for 198 of 432 accessions are not released). While I sympathize with the reasons the authors give for this, providing the sequence data that the results are based on is a standard requirement for publishing genomics papers.

Reply: The Illumina short reads analyzed by another group have already been deposited in the NCBI SRA database under the project number PRJNA766763 (<https://dataview.ncbi.nlm.nih.gov/object/PRJNA766763?reviewer=n16e951asgf3q4f1lcoe6e0u64>), and will soon be publicly available.

The SNP information for the 432 accessions, derived from the Illumina data, is available at http://solomics.agis.org.cn/potato/ftp/Genotype_432sp/. The SNP information is sufficient for sample selection for our research. We, therefore, do not damage the novelty from the other group in using the Illumina data.

Apart from this major concern, I only have a handful of minor concerns that mainly concern phrasing and statistics:

lines 147-150 "We also estimated that *Etuberosum* diverged with the common ancestor of *Lycopersicon* and *Petota* at 8.30 million years ago (MYA, 95% highest posterior: 7.9-8.8 MYA) (Supplementary Fig. 6)."

Please rephrase "highest posterior" to "highest posterior density interval", and "diverged with" to "diverged from"

Reply: We have rephrased "highest posterior" to "highest posterior density interval", and "diverged with" to "diverged from" in the revised manuscript.

lines 307-315 Still not providing evolutionary context for comparison of potatoes and tomatoes, nor establishing that genomes are of comparable quality.

Reply: In the original response to the Reviewer's comments, we have inferred the phylogeny among the 73 accessions used in this study (45 potatoes, 24 tomatoes, two *Etuberosum*, one *S. americanum* and one *S. melongena*) and estimated the divergence time (Supplementary Fig. 6). We observed that wild and cultivated tomato occurred more recently than potato. This probably suggests that the higher level of genome divergence among potato, as compared with tomato, could be due to their distinct evolutionary trajectory. Therefore, in the main text we only described the genome-wide synteny in terms of both sequence and gene levels to provide a critical implication for hybrid potato breeding, without presenting a comparison of evolutionary context of these two sections.

To indicate that the quality of those tomato genome assemblies is comparable with the potato assemblies generated in this study, we have added a description of their sequencing techniques and assembly levels in the **Methods** section in the revised manuscript.

line 502 Please rephrase "fourfold degenerated" to "fourfold degenerate"

Reply: Thanks and we have rephrased "fourfold degenerated" to "fourfold degenerate".

line 714-715 "The correlations were computed using the linear regression function "lm()" in R." Linear regression is not correlation, please check the statistics and redo analyses with `cor.test()` if required, alternatively change the methods text.

Reply: We are sorry for the confusion and have corrected the method to the function "cor()" in the revised manuscript.

lines 704-705 "To evaluate the theta value for internal branch, which reflects the level of effective population size, and thus the level of ILS, we divided the mutation units by coalescent units"
The expected degree of ILS is not only proportional to the N_e of the internal branch but also the length of that branch (i.e. timing of successive splits). Please correct this statement.

Reply: We have rephrased this sentence to "To evaluate the theta value for internal branch, which reflects the level of effective population size".

lines 871-882 on Data and code availability. All short-read data should also be deposited in a regular repository for sequence data such as NCBI.

Reply: The Illumina short reads, which are still being analyzed by another group, have already been deposited in the NCBI SRA database under the project number PRJNA766763 (<https://dataview.ncbi.nlm.nih.gov/object/PRJNA766763?reviewer=n16e951asgf3q4f1lcoe6e0u64>), and will soon be made publicly available by another research group in our institute.